atmospheric science/atmospheric chemistry

ozone layer, Proterozoic Earth, habitability, Faint Young Sun Paradox, Earth system modelling, atmospheric chemistry

**Author for correspondence:**
G. J. Cooke
e-mail: pygjc@leeds.ac.uk

# A revised lower estimate of ozone columns during Earth's oxygenated history

G. J. Cooke[1], D. R. Marsh[1,2], C. Walsh[1], B. Black[3,4] and J.-F. Lamarque[2]

[1]School of Physics and Astronomy, University of Leeds, Leeds LS2 9JT, UK
[2]National Center for Atmospheric Research, Boulder, CO 80301, USA
[3]Department of Earth and Planetary Sciences, Rutgers University, Piscataway, NJ, USA
[4]Department of Earth and Atmospheric Sciences, CUNY City College, New York, NY, USA

GJC, 0000-0001-6067-0979; DRM, 0000-0001-6699-494X;
CW, 0000-0001-6078-786X; BB, 0000-0003-4585-6438;
J-FL, 0000-0002-4225-5074

The history of molecular oxygen ($O_2$) in Earth's atmosphere is still debated; however, geological evidence supports at least two major episodes where $O_2$ increased by an order of magnitude or more: the Great Oxidation Event (GOE) and the Neoproterozoic Oxidation Event. $O_2$ concentrations have likely fluctuated (between $10^{-3}$ and 1.5 times the present atmospheric level) since the GOE ~2.4 Gyr ago, resulting in a time-varying ozone ($O_3$) layer. Using a three-dimensional chemistry-climate model, we simulate changes in $O_3$ in Earth's atmosphere since the GOE and consider the implications for surface habitability, and glaciation during the Mesoproterozoic. We find lower $O_3$ columns (reduced by up to 4.68 times for a given $O_2$ level) compared to previous work; hence, higher fluxes of biologically harmful UV radiation would have reached the surface. Reduced $O_3$ leads to enhanced tropospheric production of the hydroxyl radical (OH) which then substantially reduces the lifetime of methane ($CH_4$). We show that a $CH_4$ supported greenhouse effect during the Mesoproterozoic is highly unlikely. The reduced $O_3$ columns we simulate have important implications for astrobiological and terrestrial habitability, demonstrating the relevance of three-dimensional chemistry-climate simulations when assessing paleoclimates and the habitability of faraway worlds.

## 1. Introduction

Ozone ($O_3$), despite only making up a tiny proportion of Earth's atmosphere by weight, is one of the most important molecules for life on Earth. Without the presence of a substantial stratospheric $O_3$ layer, the surface would receive higher amounts of harmful ultraviolet (UV) radiation. However, this modern day

**Figure 1.** Earth's evolving atmosphere. (*Top*), Geochemical evidence and modelling constraints place approximate limits on the concentration of $O_2$ during this simplified history of Earth's atmospheric evolution. Brown boxes show the predicted oxygen concentration against time in the past. Grey-blue lines show approximate timelines for the appearance of the earliest life forms [6,7], the evolution of cyanobacteria [8,9], eukaryotes [10–13] and the origin of animals [14,15], with dotted lines showing a period of estimated emergence, and solid lines showing generally accepted presence. Also shown by the grey-blue square symbol is the Cambrian explosion (CE). The upward black curved arrows show approximate dates for major geological episodes of increasing atmospheric oxygenation: the Great Oxidation Event (GOE) and the Neoproterozoic Oxidation Event (NOE). Black dotted lines show the Lomagundi Event (LE) and a proposed oxygen bistability limit at 1% PAL [16]. Indicated by straight black arrows are possible 'whiffs' of increasing oxygenation. Dates and magnitude curves are not exact and there are still many uncertainties associated with several indicated events and $O_2$ constraints—see Lyons *et al.* [17], Olson *et al.* [18] and Lyons *et al.* [19]. (*Middle*), The estimated ranges for $CO_2$ and $CH_4$ are given in terms of parts per million by volume. Conflicting predictions for $CH_4$ concentrations during the Proterozoic are given by Pavlov *et al.* [20] (yellow), Olson *et al.* [21] (magenta) and Laakso & Schrag [22] (light red). The $CH_4$ (orange) and $CO_2$ (grey) ranges in the Phanerozoic and Archean are 'preferred' ranges from Olson *et al.* ([18] and references therein). The red dotted and black dotted lines are the pre-industrial values used in our simulations. (*Bottom*), The estimated luminosity of the Sun is shown with respect to time. The 'Boring Billion' is indicated, as are periods where low-latitude glaciation occurred intermittently [23,24], as shown by the dashed light blue lines next to snowflakes. Solar luminosity data are from Bahcall *et al.* [25].

$O_3$ layer would not exist without abundant molecular oxygen ($O_2$), and the Earth's atmosphere has not always been $O_2$-rich.

Atmospheres in the solar system are continuously changing, and Earth's atmosphere is no exception. From anoxic origins, Earth's atmospheric oxygenation has varied through time, with $O_2$ now the second most abundant constituent of the atmosphere. The Archean eon (4–2.4 Gyr ago) was, for the most part, a reducing atmosphere, with evidence of temporary periods of increased oxygenation [1–4]. At the end of this period, a rise in oxygen set the scene for an oxygenated biosphere and the eventual evolution of oxygen-dependent animals [5].

Figure 1 gives an overview of the current picture of Earth's oxygenation history. A large rise in oxygen concentrations occurred approximately 2.5–2.4 billion years ago at the start of the Great Oxidation Event (GOE) [19,26]. Mass-independent fractionation of sulphur isotopes in the geological record indicate that $O_2$ quantities fluctuated for a further approximately 200 Myr [26,27] before an oxygenated atmosphere was permanently established following the GOE [17,23,26,28–30]. Afterwards, oxygen concentrations dropped again [31,32], with oxygen concentrations likely between $10^{-3}$ and $10^{-1}$ the present atmospheric level (PAL equals 21% by volume, the modern day oxygen concentration) for

the rest of the Proterozoic (2.4–0.54 Gyr ago) [17,19]. Some literature estimates suggest a larger range between $10^{-5}$ and $10^{-1}$ PAL [5,18,33]. However, recent one-dimensional atmospheric photochemical modelling of Earth's oxygenation history suggests geologically persistent Proterozoic oxygen levels could have been limited to values greater than or equal to $10^{-2}$ PAL. This is based on predictions of an atmospheric bistability [16,34], where there are two stable (steady-state, converged simulations) regions of high- and trace-oxygen solutions, separated by a region where equilibrium solutions rarely exist. For instance, Gregory *et al.* [16] reported a small proportion (less than 5% of the high and trace-$O_2$ simulations) of stable solutions to exist between $3 \times 10^{-6}$% (0.6 ppmv) and 1% the present atmospheric level of $O_2$.

Towards the end of the Proterozoic, there was a further episode of increasing oxygenation known as the Neoproterozoic Oxidation Event [17,35,36], leading into the current Phanerozoic geological eon where oxygen levels have generally been estimated to have varied between 10% PAL and 150% PAL [17,33,37–40] for the past 0.54 billion years, reaching approximate modern-day concentrations during the Paleozoic [19,41,42].

The earliest fossilized animals date back approximately 575 Ma [14,43], roughly 1.7 Gyr after the GOE. Biomarkers imply that demosponges may have emerged before this, perhaps as far back as 660 Ma [44], although this has been disputed [45] and the debate continues [46–48]. Furthermore, analysis of biomolecular clocks (where rates of mutation are analysed to determine the past divergence of species and biological functionalities in lieu of alternative evidence, such as fossils) indicates possible animal life 200 Myr prior to the emergence of animal fossils [15,49]. It has been suggested that Phanerozoic-like oxygen levels were required for complex animal life (metazoans) to diversify [50–52], but not necessarily needed for metazoans to evolve [53], because the emergence of animals may not have coincided with a rise in $O_2$ [54,55]. Thus, how changing oxygen levels have influenced the evolution of life through time is uncertain [5,53,55] because of early-evolving animals such as sponges that can survive at very low $O_2$ concentrations [54].

In summary, dramatic changes in atmospheric $O_2$ levels took place during the period between approximately 2.4 and 0.4 Gyr ago, with uncertainties still covering a large $O_2$ range [19]. Previous one-dimensional modelling has found that these changes in $O_2$ strongly impact atmospheric $O_3$ levels [56–58]. In the modern atmosphere, the $O_3$ layer protects animal and plant life from harmful UV radiation, but the $O_3$ layer has not always been present. Rising oxygen levels above $10^{-4}$ PAL likely formed and increased the UV-protective $O_3$ column [e.g. see 57,58], where the column is the total number of molecules above the surface per unit area. Fluctuations in the $O_3$ column have likely affected the evolution of animal and plant life. For example, relatively rapid past reductions in the $O_3$ layer could have resulted in increased fluxes of biologically harmful UV-B radiation (280–315 nm) at the surface, possibly causing more than one mass extinction event during the Phanerozoic [59–61]. On the other hand, because Phanerozoic-like oxygen concentrations could have been present for approximately 200 Myr during the GOE [17], the protective $O_3$ column's effect on animal life's origins have been argued to be temporally irrelevant [55,62].

It is not just $O_2$ and $O_3$ concentrations that have changed since the dawn of Earth's atmosphere. The Sun's luminosity has been steadily increasing through time—see figure 1. The increased luminosity is due to the Sun fusing hydrogen into helium, which causes the central temperature and density of the Sun to increase [25], quickening the rate of fusion. The Sun's luminosity is estimated to have been 74% and 86% of today's solar luminosity, 4 Gyr ago and 2 Gyr ago, respectively [25].

The Mesoproterozoic (1.8–0.8 Gyr ago), often referred to as the 'Boring Billion', was reportedly free of widespread glaciation [24,63]. This is paradoxical, because a fainter Sun during this period would result in increased ice coverage, all other atmospheric properties being equal. This problem is the 'Proterozoic Faint Young Sun Paradox'. To mitigate the fainter Sun during various geological periods, prior research has suggested that an increased greenhouse effect is required [64–67]. Specifically for the Mesoproterozoic, elevated levels of $CO_2$ and $CH_4$ have been proposed to provide the necessary warming to avoid enhanced glaciation [20]. While higher Mesoproterozoic $CO_2$ is consistent with geological records [63], recent research has cast doubt on elevated $CH_4$ concentrations during the Mesoproterozoic due to predicted fluxes into the atmosphere that are similar to present day fluxes or lower [21,22,68]. Disregarding anthropogenic emissions, methane ($CH_4$) is currently produced on Earth primarily through biological pathways [22]. However, there are no direct indicators of $CH_4$ levels before the Pleistocene (>2.580 Myr ago) [22], so its concentration through geological time has generally been inferred through modelling. Laakso & Schrag [22] suggested Proterozoic methane levels no greater than 1 ppmv, and Olson *et al.* [21] suggested $CH_4$ concentrations were unlikely to exceed 10 ppmv, as did Daines & Lenton [68]. This is in contrast

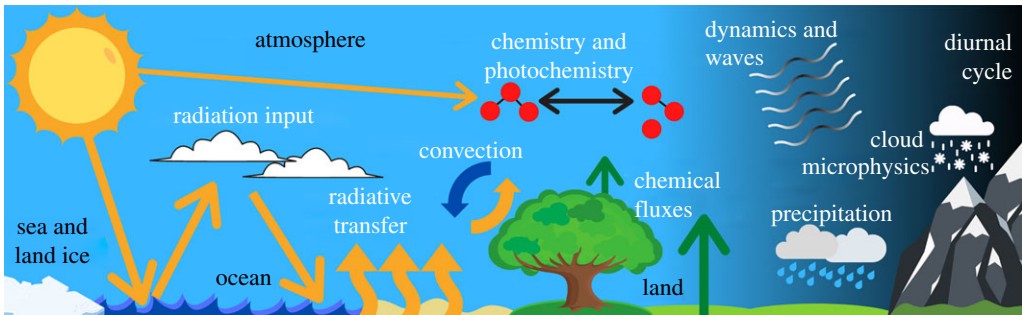

**Figure 2.** A schematic of the WACCM6 Earth System Model. In this work, WACCM6 made use of a fully interactive ocean model, as well as land-ice, sea-ice, land and atmosphere models. WACCM6 has fully coupled chemistry and physics, a state-of-the art moist physics scheme, and simulates up to roughly 140 km in altitude in the pre-industrial atmosphere.

with the suggestion of 100–300 ppmv by Pavlov *et al.* [20], which was proposed to solve this Proterozoic Faint Young Sun Paradox.

Much of the work regarding the temporal variation of the $O_3$ layer and any influence on biological habitats has been achieved through one-dimensional atmosphere modelling studies [56–58,69]. For the first time, we use a whole atmosphere chemistry-climate model to simulate three-dimensional $O_3$ variations with changing $O_2$ concentrations under Proterozoic and Phanerozoic conditions applicable to the Earth. Owing to the uncertainty in $O_2$ concentrations during these geological eons, we simulate a range of possible $O_2$ levels (0.1% PAL to a maximum of 150% PAL) since the beginning of the Proterozoic to the pre-industrial atmosphere. We demonstrate oxygen's three-dimensional influence on the $O_3$ layer (its magnitude and spatial variation) and discuss how this affects habitability (the ability for life to survive on the surface) estimates. We then determine the effects that lower $O_2$ and $O_3$ quantities have on the chemical lifetime of $CH_4$, describing how the Proterozoic Faint Young Sun Paradox is now more difficult to solve. We also determine the differences that arise when using a three-dimensional chemistry-climate model for modelling paleoclimates compared to one-dimensional modelling studies.

# 2. Atmospheric modelling using WACCM6

This work uses the most recent version of the Whole Atmosphere Community Climate Model—WACCM6 [70], which is a specific model configuration of the Community Earth System Model version 2 (CESM2).[1] A schematic of the model's capabilities is shown in figure 2. WACCM6 is a three-dimensional Earth System Model (ESM). The model couples together atmosphere, land, land-ice, ocean and sea-ice sub models. The atmosphere component in CESM2 has been updated for almost every physical regime since the previous iteration of CESM1. For instance, the code modelling moist physics and turbulence received a major update [71]. WACCM6 has 70 atmospheric layers, from a surface pressure of 1000 hPa to a pressure of $4.5 \times 10^{-6}$ hPa, the latter of which corresponds to an approximate altitude of 140 km (the lower thermosphere) for the pre-industrial atmosphere [70]. Each simulation used a horizontal grid of $2.5° \times 1.875°$ (longitude × latitude), and a 30 min time step. Previous versions of WACCM have been used for a variety of purposes, such as simulating climate change between the industrial revolution and the twenty-first century [72], as well as investigating the effects of solar flares on the middle atmosphere [73]. This same model version has also previously been used in the context of exoplanets [74–77]. Our work is the first time the WACCM configuration of CESM has been used to model the Proterozoic Earth and calculate how the $O_3$ column varies with $O_2$ concentration, although we note that Chen *et al.* [77] did simulate Proterozoic-like $O_2$ concentrations for Earth-analogue exoplanets.

We ran 12 different simulations for this work (see table 1 for a summary). Our control simulation is a pre-industrial atmosphere (hereafter PI) in which pollutants and greenhouse gas concentrations approximate those of the year 1850. This simulation starts following a 300 year control simulation with fixed 1850 conditions. We vary the mixing ratio of $O_2$ over the range of possible values during the last 2.4 billion years following the Great Oxidation Event. These levels are 150%, 50%, 10%, 5%, 1%, 0.5% and 0.1% PAL. The standard WACCM6 pre-industrial baseline simulation initial conditions were altered

---

[1]http://www.cesm.ucar.edu/models/cesm2/

**Table 1.** The 12 different simulations used for this work. There is a pre-industrial (PI) case and seven cases with varied $O_2$ levels. There are variations on the 1% PAL simulation, two with methane emissions ($CH_4$ em1 and $CH_4$ em0.1), and two with a 2 Gyr younger Sun, with pre-industrial $CO_2$ levels and four times the pre-industrial $CO_2$ levels, named YS and YS $4 \times CO_2$, respectively. The volume mixing ratio for $O_2$, $f_{O_2}$, is given in terms of present atmospheric level (PAL). The volume mixing ratio for $N_2$, $f_{N_2}$, is listed. The lower boundary condition (LBC) for $CH_4$ is shown, as well as the fixed lower boundary condition for $CO_2$ ($f_{CO_2}$). The flux of solar radiation at the top of the atmosphere, relative to today's solar constant ($S_\odot$) is given as $S$.

| simulation name | $f_{O_2}$(PAL) | $f_{N_2}$ | $CH_4$ LBC | $f_{CO_2}$ | $S$ ($S_\odot$) |
|---|---|---|---|---|---|
| PI | 1.000 | 0.78 | fixed 0.8 ppmv | 280 ppmv | 1.00 |
| 150% PAL | 1.500 | 0.68 | fixed 0.8 ppmv | 280 ppmv | 1.00 |
| 50% PAL | 0.500 | 0.89 | fixed 0.8 ppmv | 280 ppmv | 1.00 |
| 10% PAL | 0.100 | 0.97 | fixed 0.8 ppmv | 280 ppmv | 1.00 |
| 5% PAL | 0.050 | 0.98 | fixed 0.8 ppmv | 280 ppmv | 1.00 |
| 1% PAL | 0.010 | 0.98 | fixed 0.8 ppmv | 280 ppmv | 1.00 |
| $CH_4$ em1 | 0.010 | 0.98 | $5 \times 10^{14}$ g yr$^{-1}$ flux | 280 ppmv | 1.00 |
| $CH_4$ em0.1 | 0.010 | 0.98 | $5 \times 10^{13}$ g yr$^{-1}$ flux | 280 ppmv | 1.00 |
| YS | 0.010 | 0.98 | fixed 0.8 ppmv | 280 ppmv | 0.86 |
| YS $4 \times CO_2$ | 0.010 | 0.98 | fixed 0.8 ppmv | 1120 ppmv | 0.86 |
| 0.5% PAL | 0.005 | 0.98 | fixed 0.8 ppmv | 280 ppmv | 1.00 |
| 0.1% PAL | 0.001 | 0.98 | fixed 0.8 ppmv | 280 ppmv | 1.00 |

to produce each of these simulations, where the only variable change is the oxygen mixing ratio at the lower boundary—see figure 3 for the $O_2$ mixing ratio profiles. For each of these simulations, the mixing ratios of the following chemical species were held constant at the surface: $O_2$ (varied as in table 1), $CH_4$ (0.8 ppmv), $CO_2$ (280 ppmv), $N_2O$ (270 ppbv) and $H_2$ (500 ppbv). Other species, such as $O_3$ and OH were left to evolve chemically. The simulated atmospheres have a surface pressure of 1000 hPa, and in each simulation in which the mixing ratio of $O_2$ is decreased, $N_2$ is increased to maintain a 1000 hPa surface pressure. Each of these simulations uses a modern day solar spectrum.

Recent work on the bi-stability of oxygen in Earth's atmosphere suggests that oxygen levels between $3 \times 10^{-6}$% and 1% the present atmospheric level of $O_2$ are unstable on geological timescales [16]. The lowest $O_2$ concentration we simulated was 0.1% PAL. The 0.5% PAL and 0.1% PAL concentrations may not be relevant for long periods of time (that is, geologically speaking), however, this depends on oxygen's relative atmospheric flux and destruction. We note that such mixing ratios could be relevant for shorter periods of time, and such concentrations could be stable on possible exoplanet atmospheres, so we include the 0.1% PAL and 0.5% PAL simulations for this reason.

Variations on the 1% PAL simulation were also run. While some research has advocated for a methane supported greenhouse during the Proterozoic [20,78], recent research has argued that $CH_4$ in the Proterozoic atmosphere was lower, with mixing ratios similar to or lower than present day due to aqueous oxidation [21,68] or due to a low efficiency in converting organic carbon to $CH_4$ [22]. To test the impact of variable $CH_4$ concentrations, we ran a 1% PAL of $O_2$ simulation with $CH_4$ emissions where the flux of $CH_4$ to the atmosphere is the approximate modern day flux of $5 \times 10^{14}$ g yr$^{-1}$ ($CH_4$ em1), and a simulation with a reduced $CH_4$ flux of $5 \times 10^{13}$ g yr$^{-1}$ ($CH_4$ em0.1), which is based on suggested lower fluxes of $CH_4$ to the atmosphere during the Proterozoic by Laakso & Schrag [22]. We also ran two simulations using a theoretical spectrum of the Sun 2 billion years ago [79] to investigate the impact of a less luminous younger Sun. We used an existing solar evolution model [79] to produce the solar spectrum at 2 Gyr before present. The model can produce theoretical spectra for the Sun between 4.4 Gyr in the past and 3.6 Gyr in the future. The model is valid between 0.1 nm and 160 μm, and so we extend the model further into the far infrared by modelling the Sun in this region as a blackbody. The spectrum from the solar evolution model was re-binned[2] while conserving flux, to ensure that the new spectrum was interpolated onto the WACCM6 spectral irradiance grid. This young Sun's modelled total energy output was 14% less than the present Sun, with a weaker ultraviolet flux (the UV range was assumed to be between 10 nm and 400 nm) by a factor of 1.19, and

---

[2]Using a Python tool called SpectRes [80].

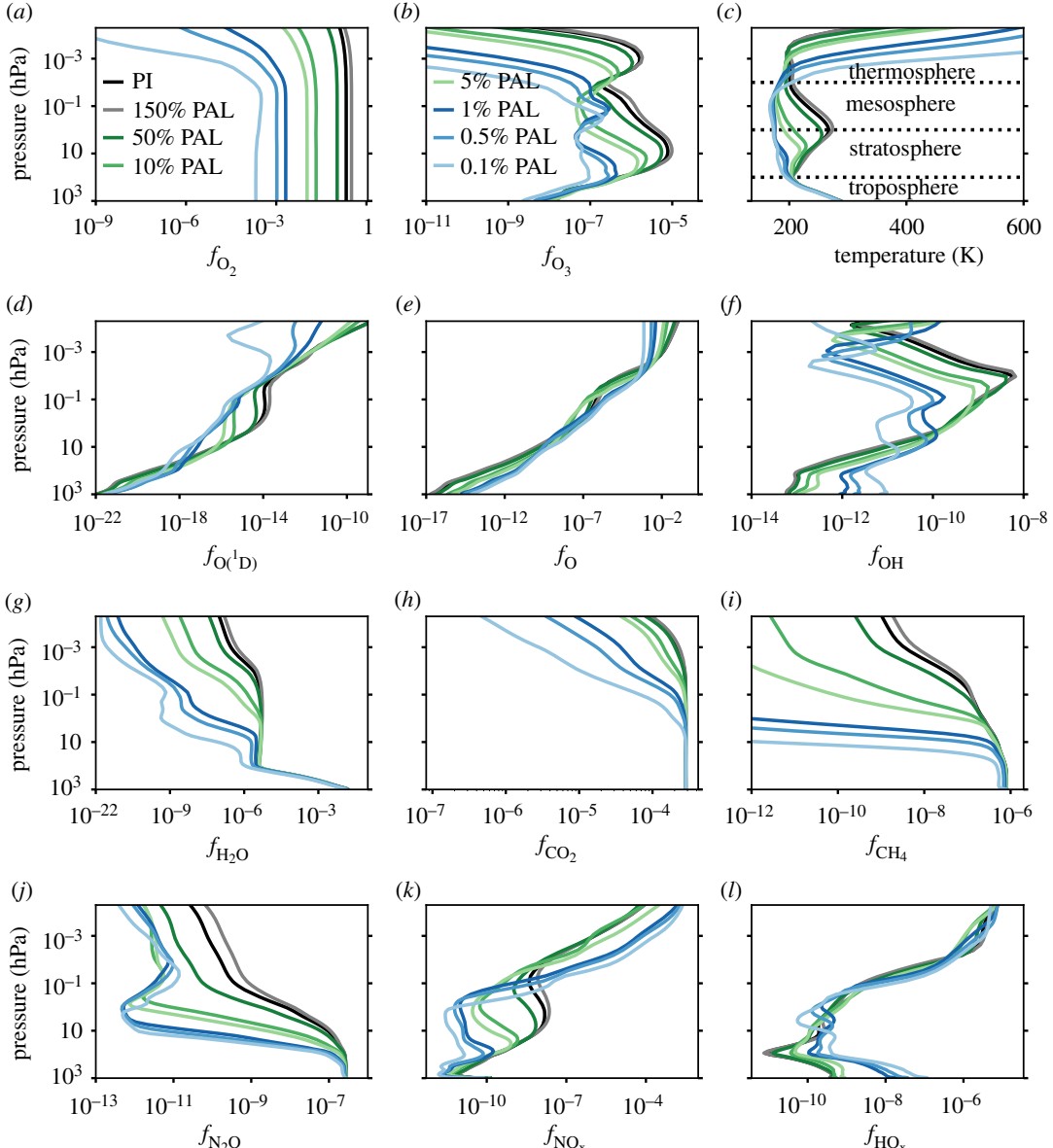

**Figure 3.** Selected time-averaged global mean atmospheric profiles output from the WACCM6 simulations are plotted. The PI (black), 150% PAL (grey), 50% PAL (dark green), 10% PAL (green), 5% PAL (light green), 1% PAL (dark blue), 0.5% PAL (blue) and 0.1% PAL (light blue) simulations are shown. PAL means relative to the present atmospheric level of $O_2$ which is 21% by volume. Mixing ratios for atmospheric constituents are shown for $O_2$ (*a*), $O_3$ (*b*), $O(^1D)$ (*d*), O (*e*), OH (*f*), $H_2O$ (*g*), $CO_2$ (*h*), $CH_4$ (*i*), $N_2O$ (*j*), NOx (*k*), and HOx (*l*). The PI atmospheric layers are indicated by black dotted lines alongside the temperature profiles in panel (*c*).

a stronger extreme ultraviolet flux (the extreme ultraviolet wavelength range was assumed to be between 10 nm and 91 nm) by a factor of 2.98. This younger Sun spectrum was used as input to the YS simulation and the YS 4 × $CO_2$ simulation. The YS simulation has a surface mixing ratio of 280 ppmv of $CO_2$ (the same as the PI simulation), while the YS 4 × $CO_2$ simulation has a surface mixing ratio of 1120 ppmv of $CO_2$, to offset the fainter Sun.

In each simulation other than the PI case, new minimum mixing ratios for $O_3$ and $CH_4$, both set at $10^{-12}$, were set to $10^{-17}$ and $10^{-25}$, respectively. A constant mixing ratio condition for $O_2$ at the lower boundary was imposed for the 0.5% and 0.1% PAL simulations because surface $O_2$ decreases below these scaled values without the imposed boundary condition. At 1% PAL and above, this does not occur on the time scales simulated.

The upper boundary conditions at $4.5 \times 10^{-6}$ hPa are even more uncertain than the lower boundary conditions because there are fewer geological proxies for the upper atmosphere. Micrometeorites have

been used to constrain the composition of the lower and upper atmosphere in the Neoarchean 2.7 Gyr ago [81–84]. For example, Tomkins *et al.* [81] and Rimmer *et al.* [82] estimated high (approx. 0.21) upper atmospheric $O_2$ concentrations, and Payne *et al.* [83] and Lehmer *et al.* [84] argued instead for high (possibly with mixing ratios of greater than 0.23) atmospheric $CO_2$ concentrations up to the homopause. Pack *et al.* [85] used micrometeorites to show that Earth's modern atmospheric $O_2$ is isotopically homogeneous below the thermosphere. Nonetheless, we do not know of any upper atmospheric constraints for the Proterozoic. We therefore ran many perturbation experiments to select upper boundary conditions in each simulation for $H_2$, H, $H_2O$, $CH_4$, O, $O_2$ and N that created smooth, consistent profiles in the thermosphere. However, we found that the upper boundary condition does not affect the atmosphere below $5 \times 10^{-5}$ hPa, as long as the upper boundary condition is not unreasonably large (for example, using a mixing ratio of 0.1 for water vapour would be unrealistic—see figure 3). The minimum pressure in our figures is thus cut to $5 \times 10^{-5}$ hPa. It is important to note that the choice of upper boundary conditions does not impact on our conclusions.

Simulations were run until the annual cycle in total hydrogen repeats for 4 years, and there were no significant surface temperature trends in the simulations where only oxygen was changed. All results presented are time-averaged means that were from the last 4 years of each simulation. Zonal means and global means are area weighted.

# 3. Results

## 3.1. The oxygen–ozone relationship

An oxygenated atmosphere enables the photochemical production of $O_3$, which is primarily produced in the tropical stratosphere, where incoming sunlight photodissociates $O_2$ and produces an oxygen atom (O). O combines with $O_2$ and any third body (M) to form $O_3$. This $O_3$ molecule can absorb ultraviolet (UV) radiation, dissociating into O and $O_2$. $O_3$ can also react with O to produce two $O_2$ molecules. This is known as the Chapman cycle [86]:

$$\left.\begin{array}{r}O_2 + h\nu \rightarrow O + O, \\ O_2 + O + M \rightarrow O_3 + M, \\ O_3 + h\nu \rightarrow O_2 + O \\ O_3 + O \rightarrow O_2 + O_2,\end{array}\right\} \tag{3.1}$$

and

where $h\nu$ represents a photon, $h$ is Planck's constant and $\nu$ is the frequency of the photon. However, the chemistry of $O_3$ is more complicated than this, with catalytic cycles involving nitrogen, hydrogen and halogen species playing an important role in destroying $O_3$ molecules [87,88]. WACCM6 includes such chemical reactions [89,90].

Figure 3 shows how imposing Proterozoic $O_2$ levels leads to striking changes in the chemical structure of the atmosphere. The maximum $O_3$ volume mixing ratio in the 0.1% PAL simulation (0.24 ppmv) is $\approx 40$ times lower than the maximum in the PI simulation (9.99 ppmv). A decrease in $O_2$ concentration results in a reduction in $O_3$ column density, which then enables increased ultraviolet flux in the lower atmosphere and increased photolysis rates. This reduces the mixing ratios of important greenhouse gases such as $H_2O$, $CH_4$, $N_2O$ and $CO_2$: from the PI simulation to the 0.1% PAL simulation, at 0.1 hPa, the time-averaged mean volume mixing ratios for these species have been reduced by factors of $8.4 \times 10^3$, $\sim 10^{18}$, 51 and 3.5, respectively.

For an atmosphere with a surface pressure of 1000 hPa, where $O_2$ has been replaced by $N_2$ to maintain the surface pressure, a greater wavelength range shortward of the visible continuum can penetrate the lower atmospheric levels. For instance, the Lyman-$\alpha$ line (121.6 nm), which is primarily absorbed by $O_2$ and usually only reaches approximately 80 km, can now photolyse $H_2O$ and $CH_4$ at lower altitudes.

Increased tropospheric photolysis of $H_2O$, given by the reaction

$$H_2O + h\nu \rightarrow OH + H, \tag{3.2}$$

and increased photolysis of $O_3$, represented by the reaction

$$O_3 + h\nu \rightarrow O(^1D) + O_2(^1D), \tag{3.3}$$

result in the production of more OH and $O(^1D)$, which are key drivers of atmospheric chemistry. OH is increased at the surface from a volume mixing ratio of $6.2 \times 10^{-14}$ to $6.9 \times 10^{-12}$ between the PI case and

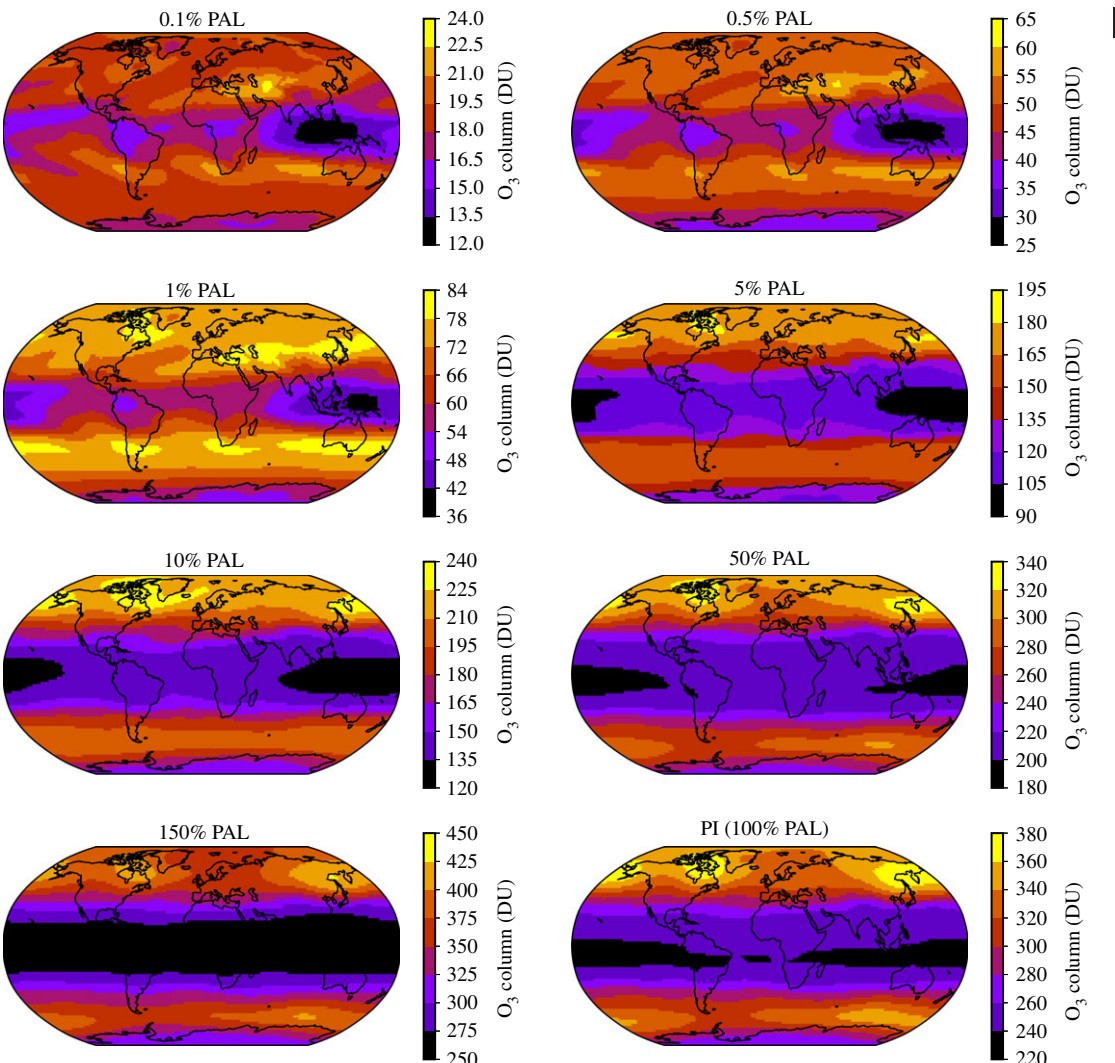

**Figure 4.** The $O_3$ column is plotted (superimposed on Earth's surface) in Dobson Units (DU) for the PI atmosphere and all the atmospheres where only oxygen concentrations were changed. Note the different scales on the colourbars. The tropics straddle either side of the equator, with the poles at the top and bottom of the two-dimensional maps, and the extratropics at intermediate latitudes.

the 0.1% PAL case. As a result of oxidation by OH and $O(^1D)$, the loss rate of $CH_4$ from the troposphere is increased by the following two reactions:

$$CH_4 + OH \rightarrow CH_3 + H_2O \tag{3.4}$$

and

$$CH_4 + O(^1D) \rightarrow CH_3 + OH. \tag{3.5}$$

There is more stratospheric HOx (HOx = H + OH + $HO_2$ + 2 · $H_2O_2$) as $O_2$ decreases which leads to further $O_3$ destruction. In the troposphere and lower stratosphere, each component of HOx is increased because the reaction

$$H_2O + h\nu \rightarrow OH + H, \tag{3.6}$$

leads to reactions that then produce more $HO_2$ and $H_2O_2$.

By contrast, NOx (N + NO + $NO_2$) is generally lower in the troposphere and stratosphere as $O_2$ is decreased. Usually, stratospheric $N_2O$ gives rise to more NOx through the reaction

$$O(^1D) + N_2O \rightarrow 2NO, \tag{3.7}$$

[91]. When $O_2$ is reduced, tropospheric and stratospheric photolysis of $N_2O$ instead produces $O(^1D)$ and $N_2$, and the path to NOx creation becomes increasingly limited with increasing photolysis.

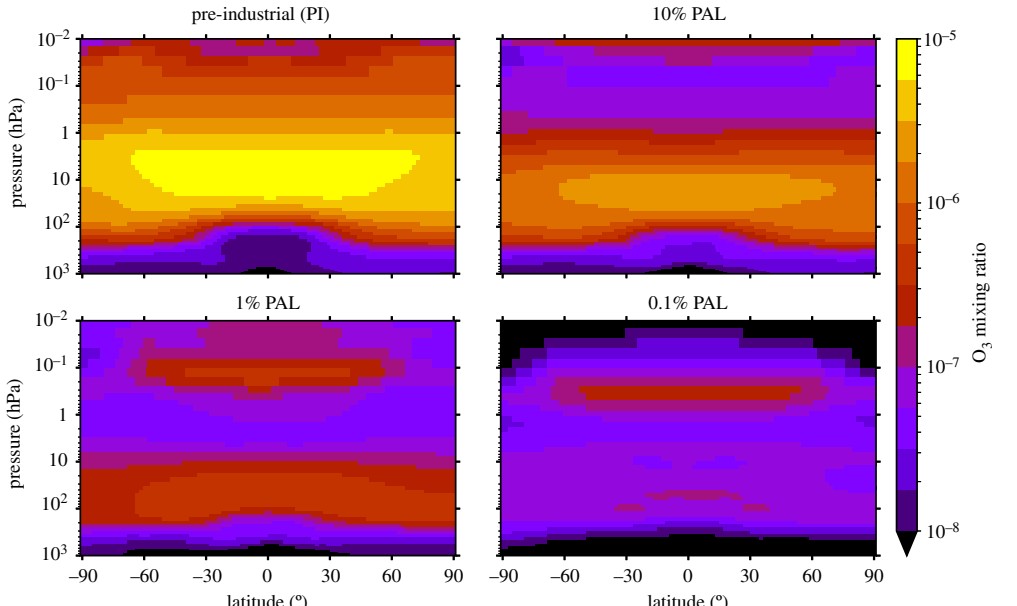

**Figure 5.** The four panels show the $O_3$ mixing ratio structure in the zonal mean (longitudinal mean) between the surface and 0.01 hPa, for the PI, 10% PAL, 1% PAL and 0.1% PAL atmospheres. The North Pole is at 90° latitude, the equator at 0°, with the South Pole at −90° latitude. The secondary night-time $O_3$ peak is not visible for the PI and 10% PAL atmospheres as it lies above 0.01 hPa.

The Earth's present-day $O_3$ column varies geographically depending on incident sunlight and the Brewer–Dobson circulation [92]. The Brewer–Dobson circulation—characterized by upwelling in the tropical stratopshere, followed by poleward movement of air parcels, then downwelling in the extratropical stratosphere—distributes $O_3$ to higher latitudes [92,93]. The $O_3$ layer thus provides varying levels of UV protection across the Earth's surface which varies with season and latitude. Figure 4 shows the annual mean geographical variation across Earth's longitudinal and latitudinal grid. The simulated global mean total $O_3$ column for the PI atmosphere case is 279 Dobson Units (1 DU = $2.687 \times 10^{20}$ molecules m$^{-2}$), decreasing to $O_3$ columns of 169 DU, 66 DU and 18 DU for the 10% PAL, 1% PAL and 0.1% PAL simulations, respectively. As $O_2$ decreases, there is a clear disruption in the pre-industrial $O_3$ distribution. Instead of the thick equatorial band of low $O_3$ levels in the PI atmosphere, the simulations which have oxygen levels ≤5% PAL have annual mean equatorial $O_3$ holes over the Pacific ocean and the Indian ocean.

Figure 5 shows the zonal mean structure of $O_3$ for the PI, 10% PAL, 1% PAL, and 0.1% PAL simulations. The stratospheric $O_3$ layer shifts in terms of altitude, shape and latitudinal variation, as does the secondary night-time $O_3$ layer. $O_3$ can be seen to trace the pressure-varying tropopause in the PI atmosphere. This is less apparent as oxygen decreases.

Displayed in figure 6 is the variation of the total $O_3$ column (and thus the modulation of surface UV fluxes) with atmospheric $O_2$ mixing ratio. The PI simulation recovers the pre-industrial $O_3$ column in both magnitude and latitudinal variation. At several $O_2$ concentrations, we report lower total $O_3$ columns compared to previous one-dimensional and three-dimensional work [56–58,69,94]. In the 10% PAL case, the mean column is approximately 1.46, 1.57, 1.76 and 2.43 times smaller when compared to Way et al. [94], Segura et al. [58], Kasting & Donahue [57] and Levine et al. [56], respectively. For the 1% PAL case, the mean $O_3$ column is approximately 1.83, 1.87, 2.24 and 2.89 times smaller when compared with Way et al. [94], Segura et al. [58], Kasting & Donahue [57] and Levine et al. [56], respectively. Also for the 1% PAL case, if we were to include minimum time-averaged values (likely at the equator where UV irradiation is highest), then the discrepancy is larger and the minimum $O_3$ column is 2.97, 3.04, 3.63 and 4.68 times smaller compared to mean $O_3$ columns from Way et al. [94], Segura et al. [58], Kasting & Donahue [57] and Levine et al. [56], respectively. We also show, along with the previous one-dimensional result from Kasting & Donahue [57] and the three-dimensional result from Way et al. [94], that $O_3$ levels consistently rise with increasing $O_2$ levels, rather than plateauing and decreasing between 0.1% PAL and 1% PAL, which previous one-dimensional models have reported [69].

The $O_3$ column is not just determined by $O_2$. It depends on many factors, including other chemical species present in the atmosphere, the flux of incoming solar radiation, and atmospheric circulation. In

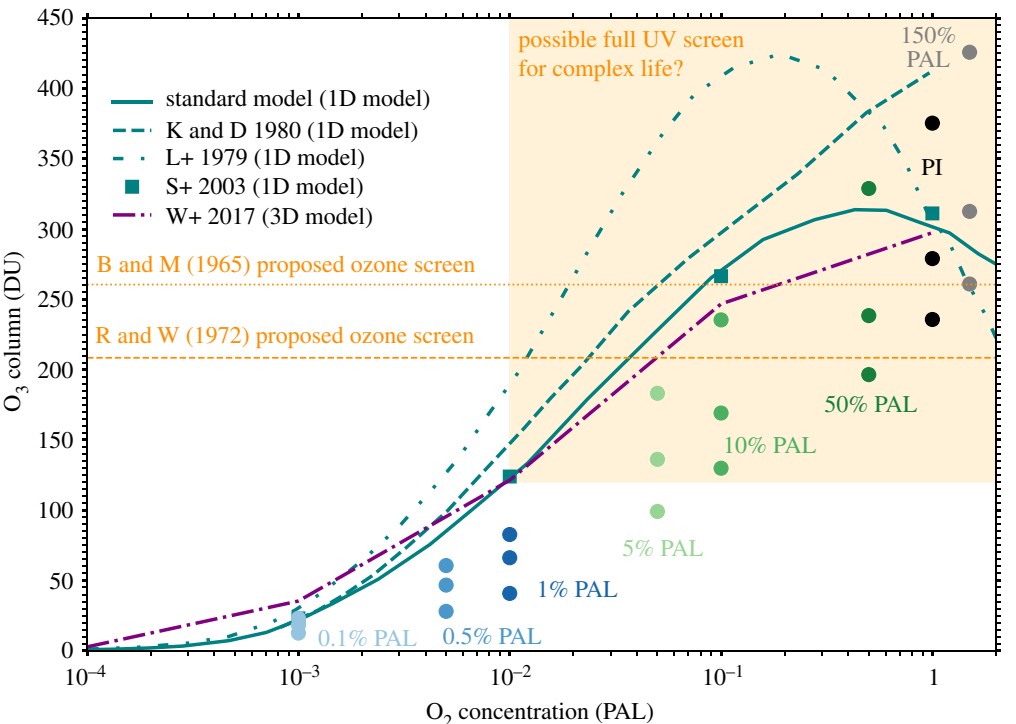

**Figure 6.** Shown by the circles (in the same colour scheme as figure 3) are the maximum, mean and minimum time-averaged $O_3$ columns from the varied $O_2$ simulations which are compared to the mean values from previous one-dimensional [56–58,69] modelling in teal, and three-dimensional [94] modelling in purple. Note that the data by Segura *et al.* [58] are indicated by the square points with no associated line. Indicated in orange shading is a proposed full UV screen, when taking into account literature assumptions (see discussion §4.1) that levels of $O_2$ at 1% PAL or higher form a fully-shielding $O_3$ layer. Also indicated by the orange dashed and orange dotted lines are the full UV shielding $O_3$ screens proposed by Berkner & Marshall [95] and Ratner & Walker [96], respectively.

figure 7, we show the impact on the $O_3$ column when we vary the $CH_4$ flux (and thus its mixing ratio), the solar spectrum, and increased $CO_2$ concentrations, in order to better simulate Proterozoic conditions. Using a spectrum of a younger Sun, which has a lower incident flux in the wavelength region that destroys $O_3$, the $O_3$ column is increased by $\approx 10$ DU. Relative to the YS simulation, the $O_3$ column in the YS $4 \times CO_2$ case is greater by $\approx 5$ DU. This is because increased $CO_2$ concentrations cool the stratosphere, mesosphere and lower thermosphere, and $O_3$ production is temperature dependent (i.e. cooler temperatures result in faster $O_3$ production). Lower $CH_4$ mixing ratios act to reduce the $O_3$ column by $\approx 5$ DU because there is more $O(^1D)$ and OH available to destroy $O_3$ that would otherwise have reacted with $CH_4$ molecules. In all cases with 1% PAL of $O_2$, the mean $O_3$ column values are lower than previous predictions.

## 3.2. The proterozoic Faint Young Sun problem

The Faint Young Sun Paradox is the problem associated with the early Sun outputting less total energy, yet the surface temperatures of Earth remaining high enough for liquid water to exist [66]. While the Faint Young Sun Paradox may have been solved for the Archean climate [67], the question of how the Earth maintained a mostly ice-free surface throughout most of the Proterozoic remains to be answered [21,63]. Some studies have suggested that an elevated $CH_4$ greenhouse can solve this problem [20,78]. In contrast with this, more recent work has suggested otherwise [21,22,68]. Here we explore possible methane concentrations during the Proterozoic.

The chemical lifetime of a molecule is its mean lifetime before it is destroyed. Reducing $O_2$ vastly reduces atmospheric chemical lifetimes for several important species, including the lifetime of $CH_4$ ($\tau_{CH_4}$)—see figure 7. The lifetime of any molecule throughout the atmosphere varies depending on photochemical destruction rates and chemical reaction rates, as well as transport of the molecule. We present global mean $\tau_{CH_4}$ profiles varying with atmospheric pressure. The actual lifetime of a $CH_4$ molecule will vary depending on where it is produced and to where it is transported.

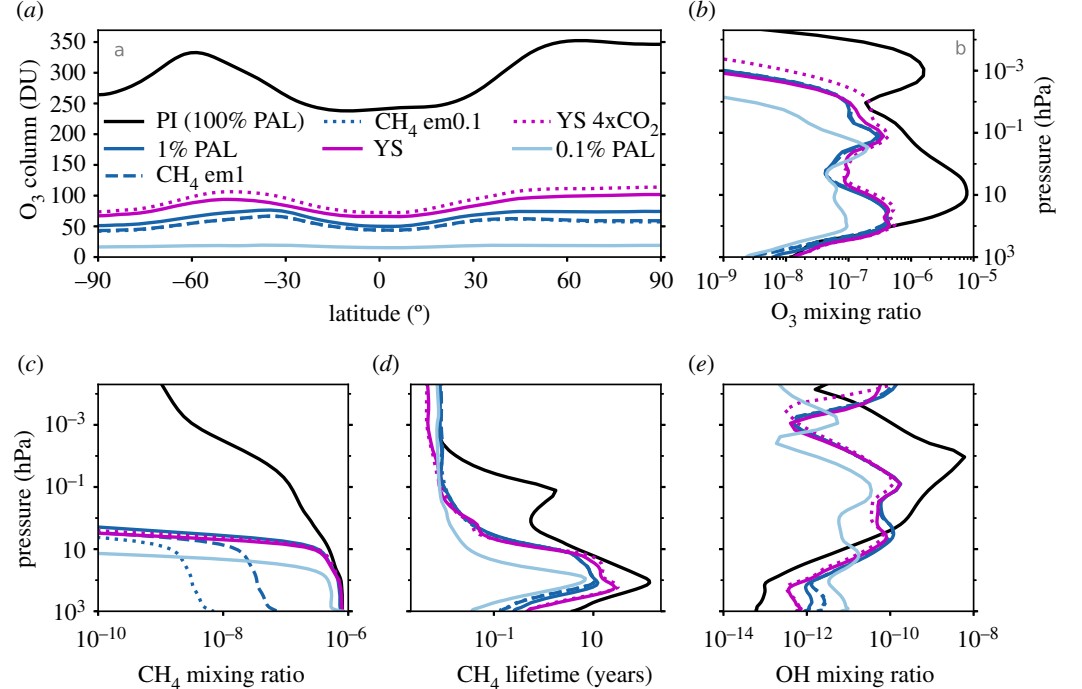

**Figure 7.** Mixing ratio and chemical lifetime of $CH_4$. In all panels, the PI (black), 1% PAL (dark blue), 0.1% PAL (light blue), $CH_4$ em1 (dark blue dashed), $CH_4$ em0.1 (dark blue dotted), YS (magenta) and YS $4 \times CO_2$ (magenta dotted) simulations are shown. The latitudinal variation of the $O_3$ column is shown in panel $a$, and the $O_3$ mixing ratio profile is shown in $b$. The $CH_4$ mixing ratio is plotted in panel $c$, and the atmospheric chemical lifetime of $CH_4$ ($\tau_{CH_4}$) is plotted in panel $d$. Note that the $CH_4$ lifetime depends on the number density of $CH_4$ and its loss rate. Finally, panel $e$ displays the OH mixing ratio.

**Table 2.** The chemical lifetime of $CH_4$ ($\tau_{CH_4}$) at the surface is given in years. This surface lifetime is then compared to the surface lifetime of $CH_4$ in the PI atmosphere ($\tau_{CH_4,PI}$) to calculate a ratio between them ($\tau_{CH_4}/\tau_{CH_4,PI}$).

| simulation name | surface $\tau_{CH_4}$ (yr) | surface $\tau_{CH_4}/\tau_{CH_4,PI}$ |
|---|---|---|
| PI | 3.813 | 1.000 |
| 150% PAL | 4.044 | 1.061 |
| 50% PAL | 3.442 | 0.903 |
| 10% PAL | 2.661 | 0.698 |
| 5% PAL | 1.983 | 0.520 |
| 1% PAL | 0.256 | 0.067 |
| $CH_4$ em1 | 0.155 | 0.041 |
| $CH_4$ em0.1 | 0.140 | 0.037 |
| YS | 0.473 | 0.124 |
| YS $4 \times CO_2$ | 0.454 | 0.119 |
| 0.5% PAL | 0.126 | 0.033 |
| 0.1% PAL | 0.037 | 0.010 |

Table 2 shows the surface $\tau_{CH_4}$ values for all the simulations. $\tau_{CH_4}$ is 3.8 years at the surface for the PI case and just 13 days for the 0.1% PAL case. The lifetime of $CH_4$ increases with height in the troposphere due to temperature-dependent chemical loss, before it decreases in the stratosphere due to photochemical loss, and oxidation with either OH or $O(^1D)$. Each of the varied $O_2$ simulations have a constant surface $CH_4$ mixing ratio. In reality, a flux to the atmosphere sustains a constant, or time-varying, surface mixing ratio. Despite using a constant mixing ratio, we can predict the flux that would be needed to sustain $CH_4$ at 0.8 ppmv in each simulation, as $\tau_{CH_4}$ does vary at the lower boundary in each case. The ratio between

the surface lifetimes is the inverse of the ratio of the surface fluxes. To illustrate, the ratio between the 1% PAL $\tau_{CH_4}$ and the PI $\tau_{CH_4}$ is 0.067, meaning that a flux increase of $1/0.067 \approx 15$ compared with the pre-industrial flux would be needed to sustain surface $CH_4$ at 0.8 ppmv in the 1% PAL simulation. All $CH_4$ lifetime ratios are given in table 2.

In the low $O_2$ simulations, a greater in magnitude increase in $CH_4$ flux, especially between 10 and 100 times the pre-industrial day flux, is unrealistic. In fact, it is possible that fluxes were lower than in the present day and could have been less than or equal to 0.1 times the present-day emissions [22]. In the $CH_4$ emissions simulations, $CH_4$ is reduced at the surface to $\approx 0.08$ ppmv and $\approx 0.007$ ppmv for the $CH_4$ em1 and $CH_4$ em0.1 simulations, respectively. So, it is likely that $CH_4$ fluxes to the atmosphere during the Proterozoic were either reduced compared to the pre-industrial flux [22], or they were not much greater [21,68], and that the atmospheric chemical lifetime of $CH_4$ was reduced due to a diminished $O_3$ column. Therefore, $CH_4$ would not have been a significant greenhouse gas during the Mesoproterozoic.

# 4. Discussion

## 4.1. Habitability and increased UV radiation

Our results show that previous one-dimensional and three-dimensional modelling may have overestimated Earth's mean $O_3$ column for atmospheric $O_2$ mixing ratios between 0.5% PAL and 50% PAL, with these mixing ratios having relevance for both the Phanerozoic and Proterozoic. In this section, we explore the potential implications for habitability during these time periods.

Assessing surface habitability is not simple. It depends on many factors, including the temperature and pressure at the surface, and also the type of life in the environment that is being evaluated. For instance, humans cannot survive in conditions where bacterial extremophiles flourish [97]. The discussion of habitability here will be limited to UV radiation, which has varying effects depending on the organism considered (note that many organisms have developed strategies to avoid excessive UV damage, as well as repair mechanisms to mitigate its effects [98–101]). Although microbial life is known to survive stronger than ambient UV irradiation [97,102,103], many animals and plant species are impacted by high doses of UV radiation, resulting in infertility [59], cell death [104] and increased mortality rates [105–107], with UV radiation considered an environmental stressor [108].

Higher surface UV fluxes during the Early Paleozoic or throughout the Proterozoic could have exerted an ecological selection pressure for organisms [109–111]. Indeed, some mass extinction events have been linked to reduced $O_3$ columns that have resulted in high UV-B fluxes [59,60,112,113]. Several decades ago, Berkner & Marshall [95] suggested that UV radiation could have prevented the colonization of dry land, but more recent literature suggests this was unlikely [55,62,102,111,114,115]. However, UV radiation may have still played a role in the subsequent evolution of life on land once it was colonized [59,60,116–118], just as stratospheric $O_3$ depletion in the last few decades, which has resulted in increased surface UV flux, has affected animals and plants in the Southern Hemisphere [119,120].

Life in the oceans experiences lower fluxes of UV radiation compared with life on land because water attenuates UV radiation [121]. There is ample evidence of life existing in the Proterozoic oceans ([11,19] and references therein), yet this does not mean that life in the photic zone (the topmost layer of the ocean which is illuminated by sunlight) would have been unaffected by UV radiation.

Photosynthesis may have been inhibited under the UV irradiance of the Proterozoic [115]. In the modern ocean, it was estimated by Smith *et al.* [122] that primary productivity[3] reduced by 6–12% under the Antarctic ozone hole. A decrease in growth rates and an increase in cell death was reported in phytoplankton by Llabrés & Agustí [123] under ambient UV-B radiation compared to no UV-B radiation. Additionally, Bancroft *et al.* [124] found through meta-analysis a widespread, overall negative effect on aquatic ecosystems from UV-B radiation, noting that the effects vary and are organism dependent. Llabrés *et al.* [125] performed a larger meta-analysis on marine biota, finding 'protists, corals, crustaceans and fish eggs and larvae' were the 'most sensitive' to increased levels of UV-B radiation. Mloszewska *et al.* [126] argued that primary productivity from cyanobacteria would have remained low until a permanent ozone screen formed at 1% PAL, citing one-dimensional modelling studies [58,127] in this assertion.

For $O_2$ concentrations between 0.5% PAL and 50% PAL, the total mean $O_3$ column quantities in our three-dimensional simulations are reduced by a factor of 1.2–2.9 times when compared to prior one-

---

[3]Primary productivity is the rate at which organic compounds are produced from $CO_2$, usually through photosynthesis.

dimensional and three-dimensional simulations [56–58,69,94]. This is maximized when considering the 1% PAL simulation minimum, with the minimum $O_3$ column reduced between 3 and 4.7 times when compared to previous mean $O_3$ column estimations. We compare the minimum here because the minimum is usually associated with the equatorial regions (figure 4), which cover a large proportion of the Earth's surface, receive the highest amounts of solar radiation, and are thus important for habitability predictions.

Whilst these reductions do not seem like large numbers, because $O_3$ reduces UV fluxes through a power law [128], an apparently small change in $O_3$ can lead to a large change in surface UV fluxes. For example, Black *et al.* [61] studied $O_3$ depletion resulting from the Siberian Traps eruptions, calculating $O_3$ columns ranging between $\approx 55$ and $\approx 145$ DU, with estimated increases in biologically damaging UV-B radiation between 5 and 50 times that of present day fluxes. Rugheimer *et al.* [129] modelled modern Earth and Earth in the past. They reported an 8.8 factor decrease in $O_3$ column (196.9 DU $\rightarrow$ 22.4 DU) between their modern Earth case and their case of Earth 2 Gyr ago. Despite total top of atmosphere UV-B (280–315 nm) and UV-C (100–280 nm) radiation decreasing 2 Gyr ago in their simulations by 1.27 and 1.29 times, respectively, this $O_3$ reduction increased biologically damaging UV fluxes by 41.3 times, with surface UV-B and UV-C fluxes increasing by a factor of 2.74 and $2 \times 10^{13}$, respectively. Segura *et al.* [58] report an $O_3$ column of 266 DU in a 10% PAL atmosphere, whilst we report a minimum of 130 DU at 10% PAL. Segura *et al.* [58] had a mean $O_3$ column of $\approx 124$ DU, but instead at 1% PAL rather than 10% PAL. For these two atmospheres (10% PAL $\rightarrow$ 1% PAL), they estimated that UV-B and UV-C surface fluxes increased by 2.08 times and 4437.5 times, respectively. The discrepancy between our simulations and prior simulations matters when estimating the habitability of a planet or exoplanet. Even the lower estimates in the literature, that suggest 0.5% PAL of $O_2$ is required to produce an effective $O_3$ screen [115], calculate UV attenuation based on $O_3$ column estimates from the one-dimensional model used by Kasting & Donahue [57]. At 0.5% PAL, our mean and minimum $O_3$ columns (45 and 30 DU) are 2.2 and 3.3 times lower than the calculated value of 100 DU by Kasting & Donahue [57].

Reasoning regarding the evolutionary impact of the $O_3$ layer and associated UV fluxes has generally been based on converged atmospheric simulations from one-dimensional models [56,57,69], which estimate that roughly 1% of the present atmospheric level of $O_2$ gives rise to an $O_3$ layer that shields the biosphere [130]. This was originally based on passing the threshold for full UV screen limits of $\approx 210$ DU proposed by Berkner & Marshall [95] and $\approx 260$ DU proposed by Ratner & Walker [96]. More recently, atmospheric [16,22], biogeochemical [36,131], biological [55,126,132], and astrobiological/ exoplanet work [74,131,133–137] have cited one-dimensional results in figure 6, often with the statement that at least 1% PAL of $O_2$ is needed to establish a full UV shield. Therefore, prior studies in figure 6 show that at 1% the present atmospheric level of $O_2$, the fully UV shielding range is between 120–185 DU for the mean $O_3$ column, whereas our 1% PAL simulation gives a mean $O_3$ column of just 66 DU, roughly half the lower end of the 120–185 DU range. Our simulations require 5% PAL of oxygen to reach a mean $O_3$ column of 136 DU, and 10% PAL to reach a mean of 169 DU and fully encompass the protective range when including our 10% PAL minimum of 130 DU. Thus, potentially 5–10 times more oxygen is required than previously thought to fully UV shield the biosphere, showing that the common assumption that 1% PAL of $O_2$ provides a full UV shield is potentially incorrect. Additionally, the real atmosphere is three-dimensional and varies temporally, and the $O_3$ layer can be influenced by biologically produced gases (e.g. $O_2$, $CH_4$), asteroid or comet impacts [138], solar activity and flares [73], as well as volcanic emissions [61].

Under reduced $O_3$ columns at $O_2$ mixing ratios between 0.5% PAL and 50% PAL, the surface and the photic zone would have received more UV radiation than previously believed. Consequently, the efficiency of photosynthesis throughout the low-$O_2$ range of the Proterozoic atmosphere could have been restricted, and UV fluxes may have acted as a stronger evolutionary variable for organisms that were susceptible to fluctuations in UV caused by $O_3$ column changes. The notion that there is a threshold above which a full-$O_3$ shield exists seems to simplify what is likely a complex interaction through Earth's oxygenated history between life's continuous evolution, and $O_2$, $O_3$ and UV radiation.

## 4.2. Origin of lower ozone columns in WACCM6

Why are our simulations predicting lower $O_3$ columns compared to previous work? Untangling the exact reasons and quantifying their magnitudes is difficult without a detailed model intercomparison. The following paragraph details possible reasons for discrepancies, which will require investigation with models to confirm.

Discrepancies with previous work may arise through the treatment of the diurnal cycle. Kasting & Donahue [57] and Segura *et al.* [58] used a solar zenith angle of 45° and multiplied photolysis rates by 0.5 to account for diurnal variation. This does not as accurately account for the temporal variation in $O_3$ throughout the atmosphere when compared to a three-dimensional model; Kasting & Donahue [57] reported that their $O_3$ profiles 'represent an upper limit on the amount of $O_3$ present at a given oxygen level'. In addition, each model will have different chemical schemes and reaction rates, which have been updated in the four decades since the work of Kasting & Donahue [57]. Another possible reason for the $O_3$ column reduction is three-dimensional transport, which is difficult to treat appropriately in one-dimensional models. The full impact of three-dimensional transport (in particular the Brewer–Dobson circulation) and the diurnal cycle treatment is uncertain. For example, Way *et al.* [94] used a three-dimensional model (ROCKE-3D) and produced lower $O_3$ columns compared to previous one-dimensional work at 10% PAL, but roughly comparative $O_3$ columns at 0.1% PAL, 1% PAL and 100% PAL. We believe that we estimate lower $O_3$ columns compared to Way *et al.* [94] because their simulations did not have fully-coupled chemistry and physics, nor did they include radiation changes directly from $O_2$ changes. To isolate the reasons for the differences, future work is required to test our hypotheses. This could include incorporating three-dimensional model chemical schemes into one-dimensional chemical schemes, and vice versa, as well as setting a constant solar zenith angle in every grid box in three-dimensional models (multiplying photolysis rates by 0.5), and potentially adjusting heating rates for constant illumination.

A minor caveat in our study is that we have not simulated $CO_2$ mixing ratios above 1120 ppmv ($4 \times$ the pre-industrial $CO_2$ mixing ratio). We note that up to 2800 ppmv of $CO_2$ may be consistent with geological proxies during the Proterozoic [18,63]. Additional $CO_2$ cooling would act to slightly increase the $O_3$ column through the temperature dependence on chemical reactions and also contribute to the absorption of Lyman-$\alpha$ radiation in simulations with the very lowest $O_2$ concentrations. Higher $CO_2$ concentrations would reduce photolysis of $H_2O$ and $CH_4$ in the upper atmosphere. However, since the absorption cross-section for $O_2$ at Lyman-$\alpha$ is $\approx 3$ times greater than that of $CO_2$ [139–141] and Lyman-$\alpha$ fluxes in the lower atmosphere would remain very small, we expect that this does not affect our new estimates of the ozone column or methane's negligible contribution to the Proterozoic greenhouse.

Our new results should not be treated as a real reconstruction of Earth's past $O_3$ states, just as our results show taking one-dimensional $O_3$ calculations as ground truth is problematic; instead, one-dimensional $O_3$ calculations should be treated with caution. The lower $O_3$ columns predicted by our work have important consequences for life's history on Earth, and the future estimation of habitability on exoplanets. At some point, Earth's atmosphere is likely to pass through varied lower oxygenated states, including analogous states to those simulated here [142]. Following Ozaki & Reinhard [142], our simulations can be used as a further step for predictions of Earth's future biosphere, its habitability, and observability. Moreover, paleoclimate modelling of the Earth that investigates specific climate events and geological processes will benefit from whole atmosphere three-dimensional chemistry-climate models that are coupled to dynamics. For instance, one-dimensional atmospheric models that investigate oxygenated exoplanet and paleo atmospheres could be tuned to replicate the lower $O_3$ column values. This tuning will also likely be applicable to oxygenated exoplanets orbiting other stellar spectral types, especially tidally locked M dwarf exoplanets, where simulating the dynamics is necessary to understand chemical transport between the day and night side of the planet.

## 4.3. Keeping the Mesoproterozoic ice-free

A reduced $O_3$ layer also affects the chemical composition of the troposphere, including the decreased abundance of $CH_4$ [21,57] caused by an increase in OH and $O(^1D)$. Methane is an important greenhouse gas, so we consider the Proterozoic greenhouse here.

The lack of evidence for glaciation during Earth's Mesoproterozoic suggests a mostly ice-free surface during this era. Given that there is ice at Earth's poles today, and during the Proterozoic there was less solar heating, then an ice-free surface without at least some increased greenhouse warming under a fainter Sun creates a contradiction, because one would expect more ice with a lower solar energy flux. To investigate this issue, we have simulated methane concentrations at varied $O_2$ concentrations and atmospheric $CH_4$ fluxes. We aim to answer the question, is it likely that a Mesoproterozoic greenhouse had substantial contributions from methane?

Three-dimensional simulations have shown that an ice-free surface can be sustained during the Mesoproterozoic if $CO_2$ is at 10 times its pre-industrial level and there is between 28 and 140 ppmv of $CH_4$ [63]. The mixing ratio of $CH_4$ at the surface in our fixed lower boundary condition simulations is

0.8 ppmv. Consequently, given the surface $\tau_{CH_4}$ values for the low $O_2$ cases, our results show that an approximate $CH_4$ flux increase (compared to present day) of a factor between 50 and 3500 is needed to reach levels of 28 ppmv during the Proterozoic (considering 10% PAL and 0.1% PAL, respectively), and five times these values to reach 140 ppmv.

Olson et al. [21] estimated that at 1% PAL of $O_2$, net biogenic $CH_4$ would be $\approx 70$ Tmol yr$^{-1}$, and the $CH_4$ mixing ratio would be at 33 ppmv. At 10% PAL of $O_2$, methane production was estimated to be closer to 20 Tmol yr$^{-1}$, with $CH_4$ concentrations of 22 ppmv (their $CH_4$ predictions vary non-linearly with $O_2$ because of further screening by the $O_3$ layer with rising $O_2$, and increased methanotrophic oxidation of $CH_4$). Laakso & Schrag [22], using a marine carbon cycling model after analysing organic carbon to $CH_4$ conversion efficiency at Lake Matano [143], calculated that for between $10^{-3}$ PAL and $10^{-1}$ PAL of oxygen during the Proterozoic, atmospheric methane mixing ratios were between 0.04 ppmv and 1 ppmv. They also estimated methane generation rates that did not exceed 50 Tmol yr$^{-1}$ (similar to the pre-industrial flux) during the Precambrian, and that Proterozoic fluxes may have been 100 times lower than this. If we were to simulate atmospheric fluxes lower than 0.5 Tmol yr$^{-1}$, then $CH_4$ surface mixing ratios would drop below 8 ppbv for 1% PAL of oxygen. A Mesoproterozoic maximum of 10% PAL $O_2$ would allow for $\approx 10 \times$ more atmospheric $CH_4$ for equivalent atmospheric fluxes, but with the Proterozoic fluxes considered here, the $CH_4$ concentration would likely not exceed 1 ppmv.

Methane fluxes remain uncertain and disputed, with huge variation in literature predictions. For example, Cadeau et al. [144] refuted the conclusions reached by Laakso & Schrag [22] after analysis of biogeochemistry in Dziani Dzaha, a volcanic crater lake with similarities to expectations of the Proterozoic oceans (e.g. it has higher salinity compared to the modern oceans). Cadeau et al. [144] concluded that methanogenesis (anaerobic methane production) resulted in efficient mineralization of the lake's high primary productivity. In this argument, Cadeau et al. [144] also cited Fakhraee et al. [145], who evaluated that Proterozoic fluxes from the oceans to the atmosphere could have been as high as 60–140 Tmol yr$^{-1}$ (9.6–22.4 $\times 10^{14}$ g yr$^{-1}$), based on predicted low-sulphate Proterozoic oceans that were mostly anoxic. Furthermore, Lambrecht et al. [146] suggested that non-diffusive transport of $CH_4$, such as the example of rising bubbles in Lake La Cruz that carry gases to the atmosphere composed of 50% $CH_4$ [147], should be considered in atmospheric models that simulate the production of $CH_4$ and its transport to the atmosphere. Regardless, in our simulations at 1% PAL of $O_2$, 140 Tmol yr$^{-1}$ (22.4 $\times 10^{14}$ g yr$^{-1}$) would not be a large enough flux to achieve $CH_4$ mixing ratios of 1 ppmv. As such, it is extremely unlikely that $CH_4$ concentrations could reach 28–140 ppmv (unless methane fluxes were larger than those found it recent literature), and thus the Mesoproterozoic could not have been kept in an ice-free state by a $CH_4$ supported greenhouse.

Could a photochemically produced haze layer prevent the reduction in methane we predict? Such a haze layer could contribute to an anti-greenhouse effect, i.e. it could cool the surface and reduce photolysis below this layer, increasing methane, which would then warm the surface. WACCM6 does not currently support the formation of organic haze from $CH_4$ photolysis, although a haze layer is unlikely to exist in our simulated atmospheres because the C/O ratio (liberated from $CH_4$ and Ox photochemistry) is $\ll 1$ in our simulations and it needs to be closer to approximately 0.5 to create a haze layer ([148,149] and references therein). This is because photolysis produces O radicals that prevent haze particle formation [148]. It has been found experimentally that haze particle production decreases as $O_2$ levels increase above $10^{-4}$ PAL, although haze particles are still produced [150]. At a pressure of $\approx 85\,000$ Pa, 0.1% PAL of $O_2$, 260 ppmv of $CO_2$, and 158 ppmv of $CH_4$, Hörst et al. [150] found a production of $\approx 1 \times 10^6$ haze particles cm$^{-3}$, such that at a C/O ratio of 0.75 (approx. 100 times greater than any C/O ratios we have simulated), these haze particles had a mixing ratio of roughly $5 \times 10^{-14}$. Furthermore, Olson et al. [21] found that hydrocarbon production from $CH_4$ photolysis during the Proterozoic would likely not result in a significant additional greenhouse contribution. When we include atmospheric $CH_4$ fluxes considering some recent estimated $CH_4$ atmospheric fluxes (table 1) [21,22], $CH_4$ concentrations are even lower than 1 ppmv, thereby further reducing the likelihood of haze formation. Owing to the fact that we consider atmospheres with a low C/O ratio and oxygen levels greater than $10^{-4}$ PAL, the effects of a haze layer are not considered important (in terms of the Proterozoic Faint Young Sun Paradox) for these reasons.

To summarize, we agree with previous work [21,22,68] after demonstrating that a Proterozoic atmosphere with a negligible $CH_4$ greenhouse contribution is more likely than one that supports a substantial $CH_4$ greenhouse. Even present day methane levels appear improbable, and this result is because of our lower $O_3$ columns (which result in further OH and O($^1$D) production—see the schematic of this in figure 8) which suggest that achieving a methane supported greenhouse during the Mesoproterozoic is even more unlikely than previously estimated. Of course, there are uncertainties in the flux of $CH_4$ during the Proterozoic, but realistic increases in $CH_4$ atmospheric flux

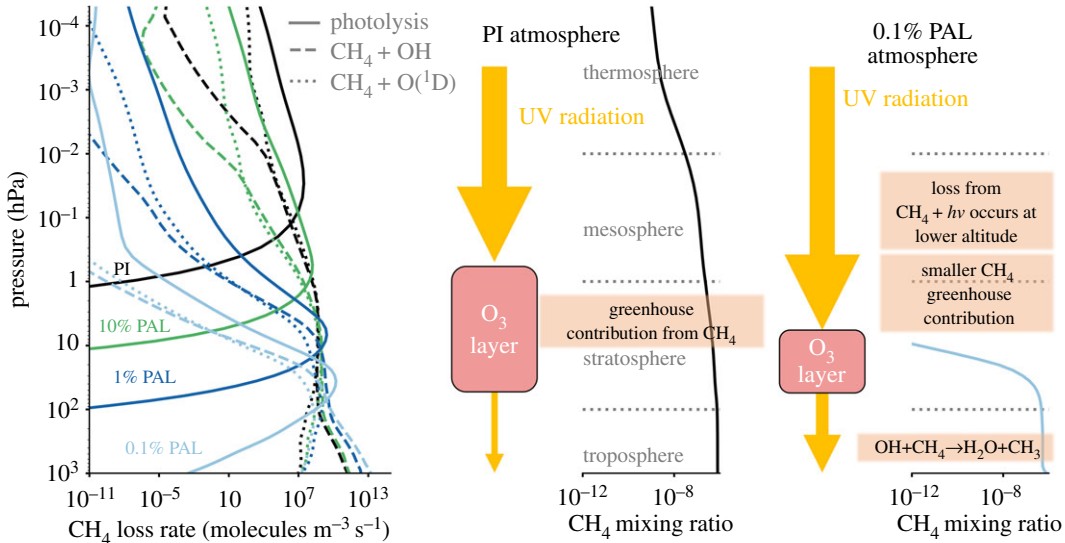

**Figure 8.** When the $O_2$ concentration reduces, the mixing ratio of $CH_4$ reduces. On the left is how the $CH_4$ loss rate varies with atmospheric pressure for the three major loss mechanisms of $CH_4$: photolysis, reaction with OH and reaction with $O(^1D)$. Shown alongside the $CH_4$ mixing ratio profiles for the PI (middle) and 0.1% PAL (right) atmospheres are yellow arrows which indicate UV radiation travelling down through the atmosphere. UV radiation is attenuated by the $O_3$ layer. When $O_2$ decreases, the $O_3$ column abundance decreases, such that increased amounts of UV radiation penetrate into the troposphere. Through photolysis, this produces more OH (e.g. photolysis of $H_2O$ and $H_2O_2$) and $O(^1D)$ molecules (e.g. photolysis of $H_2O$, $O_2$, $O_3$, and $N_2O$) which then react with $CH_4$, decreasing its abundance. $CH_4 + h\nu$ represents photolysis of $CH_4$ by a photon with frequency $\nu$, where $h$ is Planck's constant. Note that the size of the arrows and the size of the $O_3$ layers do not indicate the actual magnitude of relative UV fluxes and $O_3$ column abundances between atmospheres, respectively.

would not change the atmospheric lifetime of $CH_4$ enough to mitigate its tropospheric oxidation from OH and $O(^1D)$. Instead of a methane greenhouse, other mechanisms are required to explain a mostly ice-free Proterozoic, such as elevated levels of $N_2O$ (also unlikely due to high rates of photolysis) or $CO_2$ [63], alterations in the continental coverage [151,152], cloud variability that acts to stabilize the climate system [153], or large-scale mantle thermal mixing variations [154].

Whatever the solution, such low $CH_4$ mixing ratios have important consequences for predicted exoplanet observations that are based on Early Earth. Additionally, low $CH_4$ mixing ratios and a cool tropopause from reduced $O_3$ heating will limit the upward diffusion of hydrogen atoms to the thermosphere, with implications for atmospheric escape and exoplanetary ionospheric observations. These topics will be explored in future work.

## 5. Conclusion

We used WACCM6, a three-dimensional Earth System Model, to simulate changing oxygen levels since the beginning of the Proterozoic to a pre-industrial atmosphere. Between 0.5% and 50% the present atmospheric level of oxygen, our simulations resulted in significantly lower mean $O_3$ columns when compared to previous one-dimensional and three-dimensional modelling (figure 6). Based on common literature assumptions, we showed that between 5 and 10 times more $O_2$ is needed to produce an $O_3$ layer that fully shields the surface from biologically damaging radiation. As a consequence, we predict that UV surface fluxes were higher than previously estimated for much of Earth's history.

From these new $O_3$ column predictions, it is likely that the mixing ratio of $CH_4$ was less than $\approx 0.1$ ppmv for much of the Proterozoic. This is due to a low $CH_4$ flux to the atmosphere, as well as the increased production of tropospheric OH and $O(^1D)$ from chemical photolysis that we simulate in our model runs. As such, a methane greenhouse is unlikely to solve the Proterozoic Faint Young Sun Paradox.

$O_3$ is a crucial constituent of Earth's modern atmosphere. These results demonstrate the importance of three-dimensional whole atmosphere chemistry-climate modelling. Better constraints on Proterozoic and Phanerozoic $O_2$ levels (figure 1) will aid future work in reconstructing the history of Earth's atmosphere, the $O_3$ layer (based on our new estimates), and linking mass extinction and evolutionary events to the changing $O_3$ layer.

The $O_3$ layer varies substantially over a range of $O_2$ values, and due to its spatial variation, there were likely habitable niches across the globe as $O_2$ increased and the continents shifted. These fluctuating $O_3$ column levels through time modulated surface UV fluxes, with consequences for surface life and atmospheric chemistry. Therefore, we recommend that the biological and geological impact of the $O_3$ layer through time should be re-visited.

Data accessibility. WACCM6 is a publicly available code. The specific release used in this paper was CESM2.1.3, which can be downloaded from the following: https://escomp.github.io/CESM/versions/cesm2.1/html/downloading_cesm.html. Data are available from the Dryad Digital Repository: https://doi.org/10.5061/dryad.ncjsxksvn [155].

Authors' contributions. D.R.M., J.-F.L. and B.A.B. initiated the preliminary research. B.A.B. performed preliminary simulations. G.J.C., D.R.M. and J.-F.L. performed the final simulations. G.J.C. produced the figures. All authors analysed and interpreted the simulation output data. G.J.C. wrote the manuscript with input and comments on the final manuscript preparation from all authors.

Competing interests. We declare we have no competing interests.

Funding. G.J.C. acknowledges the studentship funded by the Science and Technology Facilities Council of the United Kingdom (STFC; grant number ST/T506230/1). C.W. acknowledges financial support from the University of Leeds, the Science and Technology Facilities Council, and UK Research and Innovation (grant numbers ST/R000549/1, ST/T000287/1, and MR/T040726/1).

Acknowledgements. We thank an anonymous reviewer, D. S. Abbot, and T. Lyons for their thorough reviews and helpful suggestions which aided in significantly improving the manuscript. Additionally, we thank J. Kasting and other anonymous reviewers for their careful reviews on a previous version of this manuscript. We thank Mark Claire and co-authors [79] for making their solar evolution model publicly available. This work was undertaken on ARC4, part of the High Performance Computing facilities at the University of Leeds, UK. We would like to acknowledge high-performance computing support from Cheyenne (doi:10.5065/D6RX99HX) provided by NCAR's Computational and Information Systems Laboratory, sponsored by the National Science Foundation. The CESM project is supported primarily by the National Science Foundation (NSF). This material is based upon work supported by the National Center for Atmospheric Research (NCAR), which is a major facility sponsored by the NSF under Cooperative Agreement 1852977.

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
