## [Peer Review File · Royal Society Open Science]

Review History

RSOS-211165.R0 (Original submission)

Review form: Reviewer 1

Is the manuscript scientifically sound in its present form?

Yes

Are the interpretations and conclusions justified by the results?

Yes

Is the language acceptable?

Yes

Do you have any ethical concerns with this paper?

No

Have you any concerns about statistical analyses in this paper?

No

Recommendation?

Accepts with minor revision (please list in comments)

Comments to the Author(s)

This is an exceptional paper that addresses a compelling issue regarding the plausible ozone columns on the Mesoproterozoic Earth. Understanding the evolution of Earth's ozone column is essential for investigating both the climatic history of the planet (through the photochemical impacts described in the study) and the likely UV radiation fluxes on the surface of the planet, which may have modulated the emergence and diversification of complex life on the surface and near surface environments. In addition, the study has implications beyond the immediate question of Mesoproterozoic ozone concentrations addressed in the paper, such as ozone column abundances on exoplanets.

Using the 3D global chemistry model WACCM to calculate ozone columns given plausible Mesoproterozoic pO_2 values is a significant leap over past studies. Those who model 1D chemistry understand and appreciate that the nature of their simulations render them approximate only and that 3D simulations are superior in terms of calculating more accurate chemical abundances for prescribed scenarios by incorporating impacts such as 3D circulations and realistic diurnal cycling. This study provides crucial results important for revising ozone column estimates through geologic time based on past studies. The authors have done an excellent job carefully describing their assumption and inputs, presenting their results and illustrating them with intuitive figures, and explaining the significance of their findings in relation to past work.

It is obvious that paper is highly polished already and I have no suggestions for major revisions. However, I do have some suggested minor revisions the authors may consider before submitting their final version. I recommend publication after consideration of these suggestions by the authors.

Minor suggestions:

Page 3, line 39, earliest animals. I believe the earliest Ediacaran fossils are actually ~575 million from the from the upper Drook Formation.

Narbonne, Guy M. "The Ediacara biota: Neoproterozoic origin of animals and their ecosystems." *Annu. Rev. Earth Planet. Sci.* 33 (2005): 421-442.

There are also putative sponge biomarkers that date to the Cryogenian (717-635 Ma), although these are very controversial.

Page 8, line 7-8, "there are fewer, if any, geologic proxies for the upper atmosphere"

Interestingly, meteorite oxidation may serve as a proxy for upper atmosphere states, see, e.g.,

Lehmer, O. R., et al. "Atmospheric CO_2 levels from 2.7 billion years ago inferred from micrometeorite oxidation." *Science advances* 6.4 (2020): eaay4644.

Payne, Rebecca C., Don Brownlee, and James F. Kasting. "Oxidized micrometeorites suggest either high pCO_2 or low pN_2 during the Neoproterozoic." *Proceedings of the National Academy of Sciences* 117.3 (2020): 1360-1366.

Page 11, line 9, it appears according to Figure 3 that the N_2O photolysis proceeds into the upper troposphere, not just the stratosphere.

Page 13, line 47, “cooler temperatures results in less O₃ production” – I think the authors mean “cooler temperatures results in *more* O₃ production”. O₃ formation is suppressed at higher temperatures. This is also the only way it makes sense in the context of the previous statement attributing CO₂ line cooling to larger O₃ columns.

In general, the study assumes a maximum CO₂ values of 4X pre-industrial concentrations. It is likely that is an underestimate (partly for the reasons elucidated in the paper). The Earth was kept warm somehow and CO₂ is the only viable contender (for many of the same reasons discussed in reference 53 that apply to the Archean). This has an impact on the context of some of the statements in the paper, such as the impact of CO₂ cooling on O₃ columns as discussed above.

In addition, on page 10 line 34-36, Lyman-alpha is also heavily shielded by CO₂. A larger CO₂ column would reduce the photolysis effect on H₂O and CH₄ reported in this section.

Page 17, line 44-47, “haze, although a haze layer is unlikely to exist in our simulated atmospheres because the CH₄/CO₂ ratio << 1 in our simulations and it needs to be closer to 0:5..” The bigger effect is that even at an O₂ level of 0.1% PAL there will be a large amount of O radicals just from O₂ and O₃ photolysis. The CH₄/CO₂ ratio is an important threshold value for anoxic atmospheres.

Review form: Reviewer 2

Is the manuscript scientifically sound in its present form?

Yes

Are the interpretations and conclusions justified by the results?

Yes

Is the language acceptable?

Yes

Do you have any ethical concerns with this paper?

No

Have you any concerns about statistical analyses in this paper?

No

Recommendation?

Accept with minor revision (please list in comments)

Comments to the Author(s)

See attached (Appendix A).

Review form: Reviewer 3

Is the manuscript scientifically sound in its present form?

Yes

Are the interpretations and conclusions justified by the results?

Yes

Is the language acceptable?

Yes

Do you have any ethical concerns with this paper?

No

Have you any concerns about statistical analyses in this paper?

No

Recommendation?

Accept with minor revision (please list in comments)

Comments to the Author(s)

See attached (Appendix B).

Decision letter (RSOS-211165.R0)

Dear Mr Cooke

The Editors assigned to your paper RSOS-211165 "A revised lower estimate of ozone columns during Earth's oxygenated history" have now received comments from reviewers and would like you to revise the paper in accordance with the reviewer comments and any comments from the Editors. Please note this decision does not guarantee eventual acceptance.

Please submit your revised manuscript and required files (see below) no later than 21 days from today's (ie 27-Oct-2021) date. Note: the ScholarOne system will 'lock' if submission of the revision is attempted 21 or more days after the deadline. If you do not think you will be able to meet this deadline please contact the editorial office immediately.

Please note article processing charges apply to papers accepted for publication in Royal Society Open Science (<https://royalsocietypublishing.org/rsos/charges>). Charges will also apply to papers transferred to the journal from other Royal Society Publishing journals, as well as papers submitted as part of our collaboration with the Royal Society of Chemistry

(<https://royalsocietypublishing.org/rsos/chemistry>). Fee waivers are available but must be requested when you submit your revision (<https://royalsocietypublishing.org/rsos/waivers>).

on behalf of Professor Peter Haynes (Subject Editor)
openscience@royalsociety.org

Associate Editor Comments to Author:

There are 3 reviews for this paper -- all very positive and making clear recommendations that the paper should be published. One reviewer has recommended what they categorise as 'moderate' revisions -- primarily concerned with the completeness/accuracy of the review material you include in the paper. To allow you a little more time to consider these recommendations carefully and either make changes or provide a brief response to why you are not making them, as appropriate, I am recommending 'major revision'. However I do not expect to send the revised paper out to referees again before acceptance.

Reviewer comments to Author:

Reviewer: 1

Comments to the Author(s)

This is an exceptional paper that addresses a compelling issue regarding the plausible ozone columns on the Mesoproterozoic Earth. Understanding the evolution of Earth's ozone column is essential for investigating both the climatic history of the planet (through the photochemical impacts described in the study) and the likely UV radiation fluxes on the surface of the planet, which may have modulated the emergence and diversification of complex life on the surface and near surface environments. In addition, the study has implications beyond the immediate question of Mesoproterozoic ozone concentrations addressed in the paper, such as ozone column abundances on exoplanets.

Using the 3D global chemistry model WACCM to calculate ozone columns given plausible Mesoproterozoic pO₂ values is a significant leap over past studies. Those who model 1D chemistry understand and appreciate that the nature of their simulations render them approximate only and that 3D simulations are superior in terms of calculating more accurate chemical abundances for prescribed scenarios by incorporating impacts such as 3D circulations and realistic diurnal cycling. This study provides crucial results important for revising ozone column estimates through geologic time based on past studies. The authors have done an excellent job carefully describing their assumption and inputs, presenting their results and illustrating them with intuitive figures, and explaining the significance of their findings in relation to past work.

It is obvious that paper is highly polished already and I have no suggestions for major revisions. However, I do have some suggested minor revisions the authors may consider before submitting their final version. I recommend publication after consideration of these suggestions by the authors.

Minor suggestions:

Page 3, line 39, earliest animals. I believe the earliest Ediacaran fossils are actually ~575 million from the from the upper Drook Formation.

Narbonne, Guy M. "The Ediacara biota: Neoproterozoic origin of animals and their ecosystems." *Annu. Rev. Earth Planet. Sci.* 33 (2005): 421-442.

There are also putative sponge biomarkers that date to the Cryogenian (717-635 Ma), although these are very controversial.

Page 8, line 7-8, "there are fewer, if any, geologic proxies for the upper atmosphere"

Interestingly, meteorite oxidation may serve as a proxy for upper atmosphere states, see, e.g.,

Lehmer, O. R., et al. "Atmospheric CO₂ levels from 2.7 billion years ago inferred from micrometeorite oxidation." *Science advances* 6.4 (2020): eaay4644.

Payne, Rebecca C., Don Brownlee, and James F. Kasting. "Oxidized micrometeorites suggest either high pCO₂ or low pN₂ during the Neoproterozoic." *Proceedings of the National Academy of Sciences* 117.3 (2020): 1360-1366.

Page 11, line 9, it appears according to Figure 3 that the N₂O photolysis proceeds into the upper troposphere, not just the stratosphere.

Page 13, line 47, "cooler temperatures results in less O₃ production" – I think the authors mean "cooler temperatures results in *more* O₃ production". O₃ formation is suppressed at higher temperatures. This is also the only way it makes sense in the context of the previous statement attributing CO₂ line cooling to larger O₃ columns.

In general, the study assumes a maximum CO₂ values of 4X pre-industrial concentrations. It is likely that is an underestimate (partly for the reasons elucidated in the paper). The Earth was kept warm somehow and CO₂ is the only viable contender (for many of the same reasons discussed in reference 53 that apply to the Archean). This has an impact on the context of some of the statements in the paper, such as the impact of CO₂ cooling on O₃ columns as discussed above.

In addition, on page 10 line 34-36, Lyman-alpha is also heavily shielded by CO₂. A larger CO₂ column would reduce the photolysis effect on H₂O and CH₄ reported in this section.

Page 17, line 44-47, "haze, although a haze layer is unlikely to exist in our simulated atmospheres because the CH₄/CO₂ ratio << 1 in our simulations and it needs to be closer to 0:5.." The bigger effect is that even at an O₂ level of 0.1% PAL there will be a large amount of O radicals just from O₂ and O₃ photolysis. The CH₄/CO₂ ratio is an important threshold value for anoxic atmospheres.

Reviewer: 2

Comments to the Author(s)

See attached ("review.pdf").

Reviewer: 3

Comments to the Author(s)

See attached:

- RS.pdf
- sciadv.1600134.pdf
- Lambrecht_et_al-2019-Geobiology copy.pdf
- supp_data copy.pdf
- ast.2020.2418 copy.pdf

===PREPARING YOUR MANUSCRIPT===

===PREPARING YOUR REVISION IN SCHOLARONE===

Author's Response to Decision Letter for (RSOS-211165.R0)

See Appendix C.

Decision letter (RSOS-211165.R1)

Dear Mr Cooke,

It is a pleasure to accept your manuscript entitled "A revised lower estimate of ozone columns during Earth's oxygenated history" in its current form for publication in Royal Society Open Science. The comments of the reviewer(s) who reviewed your manuscript are included at the foot of this letter.

[The previous editorial decision was to enable you to respond thoroughly to the referees' comments -- which you have done, therefore there is no reason to delay publication of the paper further and I am pleased to accept it.]

The proof of your paper will be available for review using the Royal Society online proofing system and you will receive details of how to access this in the near future from our production office (opencscience_proofs@royalsociety.org). We aim to maintain rapid times to publication after acceptance of your manuscript and we would ask you to please contact both the production office and editorial office if you are likely to be away from e-mail contact to minimise delays to publication. If you are going to be away, please nominate a co-author (if available) to manage the proofing process, and ensure they are copied into your email to the journal.

Kind regards,
Royal Society Open Science Editorial Office
Royal Society Open Science
opencscience@royalsociety.org

on behalf of Professor Peter Haynes (Subject Editor)
openscience@royalsociety.org

Appendix A

Paper: A revised lower estimate of ozone columns during Earth's oxygenated history

Authors: Cooke et al.

Journal: RSOS

Reviewer: Dorian S. Abbot

Date: October 8, 2021

Overview: This is a fascinating paper that presents extremely interesting new 3D coupled climate-chemistry simulations of Earth's atmosphere at different O_2 levels using the WACCM6 Earth System Model. It is important to emphasize that these simulations are computationally expensive, take a long time to perform, produce a huge amount of data that must be carefully organized and stored and take lots of time and expertise to analyze. The authors have done an excellent job at all of this, and should be proud of their significant contribution to the literature. As far as I know, they are presenting the first 3D coupled climate-chemistry simulations relevant for the problem under consideration. This paper definitely deserves to be published quickly. My comments concern questions the paper raised for me that the authors may be able to address with their data archives and a few new simulations.

Major Comments:

1. Comparison with other work: A major conclusion of this paper is that O_3 columns are lower than previous studies have found for a given O_2 level. This result is presented in Figure 6 and page 11, as well as discussed on page 16. After reading this I was left unsatisfied about why this is the case. It's worth thinking through the specific mechanistic explanation and presenting it. The fact that WACCM6 is newer and includes more processes does not necessarily mean it is correct. The reader will have more confidence in your results if you can explain the exact improvement that leads to the difference. Here are some suggestions for working on this:

1. Make it more clear in Figure 6 and in the text which previous simulations are 1D and which are 3D.
2. Perform 1D simulations with the WACCM6 chemical core and compare them to previous 1D work. This will allow you to separate out differences in chemical schemes from effects of going from 1D to 3D.
3. Perform simulations in WACCM6 with chemistry and physics decoupled. Test if this is the main reason for differences with Michael Way's ROCKE3D results, or if it is due to a different chemical scheme or differences in the radiative effect of O_2 .
4. You point to the diurnal cycle as a key to differences with previous work. This is easy enough to test. When you have a 1D version of WACCM6 chemistry you can turn a diurnal cycle on and off and see what effect it has. You can also do this in the full 3D version, but this will be more expensive.

2. Methane: I am not an atmospheric chemist, but I was surprised by the result that less O_2 leads to less CH_4 (if I understood correctly). I would have assumed that increasing O_2 increases oxidation

of CH_4 and would therefore decrease its equilibrium value. I think it would be good to specifically contrast your results with this intuition that many people may have. I also think it would be good to explain how this works in a bit more detail and add a schematic figure showing it. If others are naive like me about this, it could be a good way to educate the field.

Appendix B

This paper, at its core, is a numerical simulation of atmospheric ozone distributions across Earth history based on previous, wide-ranging estimates for early O₂ levels. Those estimates are particularly uncertain for the Proterozoic, which the authors fully acknowledge. As a result, they explore the range of possibilities for atmospheric O₂ presented within the literature. The estimates for ozone are then discussed within the framework of surface habitability (in terms of associated UV fluxes), methane stability in the atmosphere (in terms of predicted photochemistry), and climate (in terms of possible greenhouse gas scenarios).

While the authors are not the first to explore O₂-O₃-CH₄ space for the Precambrian, the uncertainties in such work remain high, and fresh perspectives/modeling efforts are always welcome. In this regard, the authors could highlight the novelty of their work even more, which to me is the bridging of the modeling and the very recent discussions about O₂ variability and patterns of life and climate. In other words, the models are informed by current estimates and debates for atmospheric O₂ evolution and so are very timely. Also, novel/unique details of their model approach relative to previous work could be clearer.

I strongly support publication following moderate revision. For that revision, the paper has a lot of review information, including fairly extensive treatments of aspects of the history of life, which are not always well tied (specifically relevant) to their particular contribution about ozone and are not always fully up to date.

Some suggestions follow:

- (1) The attached Lyons et al. paper might be helpful for some details, particularly discussions about the history of life, atmospheric O₂ estimates, possible controls on mid-Proterozoic ocean/atmosphere O₂, and the NOE. No pressure for extensive inclusion, but it could be a useful resource during revision.
- (2) There are discussions about UV fluxes and potential impacts of life on land, including animals (p. 14), which is not really the focus of the cited ref. 30. Are there really doubts about sufficient ozone shielding at the time that animals first invaded land well into the Paleozoic? This aspect of the impacts on life could be discussed more thoroughly or removed.
- (3) Similarly, while low O₃-high UV are issues for life on land, they would be less so for life in the oceans, even shallow oceans. As such, the relationship between UV shielding and life in the Proterozoic oceans should be addressed. We know that those oceans were teeming with prokaryotic life and emerging eukaryotes. It's not clear to me that there is a relationship between those patterns of life and UV fluxes, although we do commonly talk about rising O₂ as driving eukaryotic diversification and ultimately the emergence of animals. But I tend to think about this in terms of respiratory O₂ requirements rather than UV fluxes.
- (4) The authors discuss methane possibilities within the context of fluxes estimates by Laakso and Schrag (2019) and Olson et al. (2016). That's great, but the details of those papers have been challenged based on possibilities for methane transport beyond what is possible by diffusion alone (e.g., bubbles). I encourage the authors

to address this caveat as they decide what methane levels to assume (and to see Lambrecht et al., attached). It's something to mention.

- (5) This will seem like nitpicking, but some of the details in the introduction are incorrect and/or confusing. First, the loss of S-MIF was not 2.5 to 2.4 billion years ago and is more complicated than the authors suggest. I encourage them to have a look at Luo et al. (attached) and Poulton et al. (2021, cited by the authors).
- (6) The early animal details in the intro are not very rigorous, including an absence of discussion about the earliest records (biomarkers), which well predate the fossils the authors discuss (Lyons et al. could be useful here too).
- (7) And certainly most folks do not think that oxygen rose to near-modern levels by 550 million years ago. There is evidence, for example, for a persistence of low oxygen in the oceans well into the Paleozoic, although specifics for the atmosphere are harder to constrain. There is a lot of recent literature on these stories (Lyons et al., attached, the supplement in particular, could help with some of this).
- (8) P. 2 (Line 24): Where does the estimate of 10^{-4} come from? That is very low.
- (9) P. 2 (Lines 27-29) I am confused by the assertion that 1% PAL is higher than 10^{-1} (10%) PAL. Be careful when switching between fraction and percent PAL. In general, this intro is good but not great and could be ramped up and trimmed of pieces not immediately tied to the authors' work/conclusions.

Overall, nice stuff. Should definitely be published.

Appendix C

Response to Reviewers (RSOS-211165)

We appreciate the time all the reviewers have taken to carefully read our submitted manuscript, and we would like to thank them for their feedback, comments, and suggestions, which we believe have helped to improve the manuscript. We have responded to all the comments from the reviewers, and we have made changes to the manuscript accordingly. Reviewer points are in bold, and our responses to these points are in red.

Response to reviewer 1:

- 1. Page 3, line 39, earliest animals. I believe the earliest Ediacaran fossils are actually ~575 million from the from the upper Drook Formation.**

Narbonne, Guy M. "The Ediacara biota: Neoproterozoic origin of animals and their ecosystems." *Annu. Rev. Earth Planet. Sci.* 33 (2005): 421-442.

There are also putative sponge biomarkers that date to the Cryogenian (717-635 Ma), although these are very controversial.

We changed the age from ~550 Myr ago to ~575 Myr ago, including the Narbonne, Guy M (2005) reference given by the reviewer. The sentence now reads:

'The earliest fossilised animals date back ~ 575 Myr ago [28, 29], roughly 1.7 Gyr after the GOE.'

We include the reference to biomarkers in the sentence afterwards:

'Biomarkers imply that demosponges may have emerged before this, perhaps as far back as 660 Myr ago [30], although this has been disputed [31] and the debate continues [32, 33, 34].'

- 2. Re: Page 8, line 7-8, "there are fewer, if any, geologic proxies for the upper atmosphere"**

Interestingly, meteorite oxidation may serve as a proxy for upper atmosphere states, see, e.g.,

Lehmer, O. R., et al. "Atmospheric CO₂ levels from 2.7 billion years ago inferred from micrometeorite oxidation." *Science advances* 6.4 (2020): eaay4644.

Payne, Rebecca C., Don Brownlee, and James F. Kasting. "Oxidized micrometeorites suggest either high pCO₂ or low pN₂ during the Neoproterozoic." *Proceedings of the National Academy of Sciences* 117.3 (2020): 1360-1366.

We have changed the beginning of this paragraph, including the references the reviewer mentioned which are [86,87], as well as additional references [84,85,88] relevant to geological proxies from micrometeorites:

'The upper boundary conditions at 4.5×10^{-6} hPa are even more uncertain than the lower boundary conditions because there are fewer geological proxies for the upper atmosphere. Micrometeorites have been used to constrain the composition of the lower and upper atmosphere in the Neoproterozoic 2.7 Gyr ago [81, 82, 83, 84]. For example, Tomkins et al. [81] and Rimmer et al. [82] estimated high (~ 0.21) upper atmospheric O_2 concentrations, and Payne et al. [83] and Lehmer et al. [84] argued instead for high (possibly with mixing ratios of > 0.23) atmospheric CO_2 concentrations up to the homopause. Pack et al. [85] used micrometeorites to show that Earth's modern atmospheric O_2 is isotopically homogeneous below the thermosphere. Nonetheless, we do not know of any upper atmospheric constraints for the Proterozoic.'

3. **Re: Page 11, line 9, it appears according to Figure 3 that the N_2O photolysis proceeds into the upper troposphere, not just the stratosphere.**

The reviewer is correct, and we have changed the sentence from 'When O_2 is reduced, stratospheric photolysis of N_2O instead produces $O(^1D)$ and $N_2...$ ' to 'When O_2 is reduced, **tropospheric and** stratospheric photolysis of N_2O instead produces $O(^1D)$ and $N_2...$ '

4. **Re: Page 13, line 47, "cooler temperatures results in less O_3 production" – I think the authors mean "cooler temperatures results in *more* O_3 production". O_3 formation is suppressed at higher temperatures. This is also the only way it makes sense in the context of the previous statement attributing CO_2 line cooling to larger O_3 columns.**

The reviewer is correct that this was a typo. We have corrected the statement to instead say 'cooler temperatures result in **faster** O_3 production'

5. **Re: In general, the study assumes a maximum CO_2 values of 4X pre-industrial concentrations. It is likely that is an underestimate (partly for the reasons elucidated in the paper). The Earth was kept warm somehow and CO_2 is the only viable contender (for many of the same reasons discussed in reference 53 that apply to the Archean). This has an impact on the context of some of the statements in the paper, such as the impact of CO_2 cooling on O_3 columns as discussed above.**

In addition, on page 10 line 34-36, Lyman-alpha is also heavily shielded by CO_2 . A larger CO_2 column would reduce the photolysis effect on H_2O and CH_4 reported in this section.

We have noted that higher CO_2 values during the Proterozoic are consistent with geological proxies. We have also noted that greater CO_2 levels will further affect the O_3 column and Lyman- α radiation. We have updated the discussion (section 4b) with the following paragraph to reflect this:

'A minor caveat in our study is that we have not simulated CO_2 mixing ratios above 1120 ppmv (4x the pre-industrial CO_2 mixing ratio). We note that up to 2800 ppmv of CO_2 may be consistent with geological proxies during the Proterozoic [16, 51]. Additional CO_2 cooling would act to slightly increase the O_3 column through the temperature dependence on chemical reactions and also contribute to the absorption of Lyman- α radiation in simulations with the very lowest O_2 concentrations. Higher CO_2 concentrations would

reduce photolysis of H₂O and CH₄ in the upper atmosphere. However, since the absorption cross section for O₂ at Lyman-α is ≈ 3 times greater than that of CO₂ [139, 140, 141] and Lyman-α fluxes in the lower atmosphere would remain very small, we expect that this does not affect our new estimates of the ozone column or methane's negligible contribution to the Proterozoic greenhouse.'

6. **Re: Page 17, line 44-47, "haze, although a haze layer is unlikely to exist in our simulated atmospheres because the CH₄/CO₂ ratio << 1 in our simulations and it needs to be closer to 0:5.." The bigger effect is that even at an O₂ level of 0.1% PAL there will be a large amount of O radicals just from O₂ and O₃ photolysis. The CH₄/CO₂ ratio is an important threshold value for anoxic atmospheres.**

We thank the reviewer for pointing this out, and we have amended the text in the paragraph as a result of this (we have C/O ratio in the text now instead of CH₄/CO₂ ratio)

'WACCM6 does not currently support the formation of organic haze from CH₄ photolysis, although a haze layer is unlikely to exist in our simulated atmospheres because the C/O ratio (liberated from CH₄ and Ox photochemistry) is << 1 in our simulations and it needs to be closer to ~ 0.5 to create a haze layer [148, 149, and references therein].'

Response to T. Lyons:

1. The attached Lyons et al. paper might be helpful for some details, particularly discussions about the history of life, atmospheric O₂ estimates, possible controls on mid-Proterozoic ocean/atmosphere O₂, and the NOE. No pressure for extensive inclusion, but it could be a useful resource during revision.

We thank the reviewer for pointing us towards Lyons et al 2021, and we have modified our manuscript in accordance with some of the reviewer's suggestions below.

2. There are discussions about UV fluxes and potential impacts of life on land, including animals (p. 14), which is not really the focus of the cited ref. 30. Are there really doubts about sufficient ozone shielding at the time that animals first invaded land well into the Paleozoic? This aspect of the impacts on life could be discussed more thoroughly or removed.

ref. 30 was Daniel B Mills and Donald E Canfield. Oxygen and animal evolution: Did a rise of atmospheric oxygen "trigger" the origin of animals? BioEssays, 36(12):1145–1155, 2014, doi = <https://doi.org/10.1002/bies.201400101>.

Whilst UV radiation is not the main focus of the paper, Box 1 in Mills and Canfield (2014) present ozone UV shielding arguments that are based on prior 1D modelling studies that are between 18-50 years old. Our discussion aims to highlight many similar assumptions in the literature when viewed in the context of our updated O₃ columns.

Nonetheless, we agree that the discussion could be more developed. We have removed the following paragraph:

'Given that our O₃ column calculations are lower than previous estimations by significant amounts, and surface land may or may not be associated with O₃ column regional maxima in the past, the O₃ layer could have played a more important evolutionary role than has recently been considered for life on land, especially for complex life such as animals and plants.'

We have re-ordered some of the discussion section, as well as adding some more points (new text highlighted in bold):

'Higher surface UV fluxes during the early Paleozoic or throughout the Proterozoic could have exerted an ecological selection pressure for organisms [109, 110, 111]. Indeed, some mass extinction events have been linked to reduced O₃ columns that have resulted in high UVB fluxes [46, 47, 112, 113]. Several decades ago, Berkner and Marshall [92] suggested that UV radiation could have prevented the colonisation of dry land, but more recent literature suggests this was unlikely [42, 49, 102, 111, 114, 115]. However, UV radiation may have still played a role in the subsequent evolution of life on land once it was colonised [46, 47, 116, 117, 118], just depletion in the last few decades, which has resulted in increased UV flux, has affected animals and plants in the southern hemisphere [119, 120].'

3. Similarly, while low O₃-high UV are issues for life on land, they would be less so for life in the oceans, even shallow oceans. As such, the relationship between UV shielding and life in the Proterozoic oceans should be addressed. We know that those oceans were teeming with prokaryotic life and emerging eukaryotes. It's not clear to me that there is a relationship between those patterns of life and UV fluxes, although we do commonly talk about rising O₂ as driving eukaryotic diversification and ultimately the emergence of animals. But I tend to think about this in terms of respiratory O₂ requirements rather than UV fluxes.

We thank the reviewer for pointing this out. We have addressed the fact that UV fluxes will have been attenuated in the oceans, adding significant discussion to the effect of UV on life in the oceans.

'Life in the oceans experiences lower fluxes of UV radiation compared to life on land because water attenuates UV radiation [121]. There is ample evidence of life existing in the Proterozoic oceans [6, 66, and references therein], yet this does not mean that life in the photic zone (the topmost layer of the ocean which is illuminated by sunlight) would have been unaffected by UV radiation.'

Photosynthesis may have been inhibited under the UV irradiance of the Proterozoic [115]. In the modern ocean, it was estimated by Smith et al. [122] that primary productivity³ reduced by 6-12% under the Antarctic ozone hole. A decrease in growth rates and an increase in cell death was reported in phytoplankton by Llabrés and Agustí [123] under ambient UV-B radiation compared to no UV-B radiation. Additionally, Bancroft et al. [124] found through meta-analysis a widespread, overall negative effect on aquatic ecosystems from UV-B radiation, noting that the effects vary and are organism dependent. Llabrés et al. [125] performed a larger meta-analysis on marine biota, finding 'protists, corals, crustaceans and fish eggs and larvae' were the 'most sensitive' to increased levels of UV-B radiation. Mloszewska et al. [126] argued that primary productivity from cyanobacteria

would have remained low until a permanent ozone screen formed at 1% PAL, citing 1D modelling studies [45, 127] in this assertion.'

4. The authors discuss methane possibilities within the context of fluxes estimates by Laakso and Schrag (2019) and Olson et al. (2016). That's great, but the details of those papers have been challenged based on possibilities for methane transport beyond what is possible by diffusion alone (e.g., bubbles). I encourage the authors to address this caveat as they decide what methane levels to assume (and to see Lambrecht et al., attached). It's something to mention.

We have addressed this caveat with the reference the reviewer suggested, and also mentioned alternative CH₄ fluxes to the atmosphere to broaden the discussion. We have made the point that this issue remains unresolved.

'Methane fluxes remain uncertain and disputed with huge variation in literature predictions. For example, Cadeau et al. [144] refuted the conclusions reached by Laakso and Schrag [59] after analysis of biogeochemistry in Dziani Dzaha, a volcanic crater lake with similarities to expectations of the Proterozoic oceans (e.g. it has higher salinity compared to the modern oceans). Cadeau et al. [144] concluded that methanogenesis (anaerobic methane production) resulted in efficient mineralisation of the lake's high primary productivity. In this argument, Cadeau et al. [144] also cited Fakhraee et al. [145], who evaluated that Proterozoic fluxes from the oceans to the atmosphere could have been as high as 60 to 140 Tmol yr⁻¹ (9.6 – 22.4 × 10¹⁴ g yr⁻¹), based on predicted low-sulphate Proterozoic oceans that were mostly anoxic. Furthermore, Lambrecht et al. [146] suggested that non-diffusive transport of CH₄, such as the example of rising bubbles in Lake La Cruz that carry gases to the atmosphere composed of 50% CH₄ [147], should be considered in atmospheric models that simulate the production of CH₄ and its transport to the atmosphere. Regardless, in our simulations at 1% PAL of O₂, 140 Tmol yr⁻¹ (22.4 × 10¹⁴ g yr⁻¹) would not be a large enough flux to achieve CH₄ mixing ratios of 1 ppmv. As such, it is extremely unlikely that CH₄ concentrations could reach 28-140 ppmv (unless methane fluxes were larger than those found in recent literature), and thus the Mesoproterozoic could not have been kept in an ice-free state by a CH₄ supported greenhouse.'

5. This will seem like nitpicking, but some of the details in the introduction are incorrect and/or confusing. First, the loss of S-MIF was not 2.5 to 2.4 billion years ago and is more complicated than the authors suggest. I encourage them to have a look at Luo et al. (attached) and Poulton et al. (2021, cited by the authors).

We have changed the paragraph indicated by the reviewer to reflect the fact that S-MIF did not disappear entirely at the start of the GOE as we originally stated.

'A large rise in oxygen concentrations occurred approximately 2.5-2.4 billion years ago at the start of the Great Oxidation Event [GOE; 6, 7]. Mass-independent fractionation of sulphur isotopes in the geological record indicate that O₂ quantities fluctuated for a further ~200 million more years [7, 8] before an oxygenated atmosphere was permanently established following the GOE [7, 9, 10, 11, 12, 13].'

6. The early animal details in the intro are not very rigorous, including an absence of discussion about the earliest records (biomarkers), which well predate the fossils the authors discuss (Lyons et al. could be useful here too).

As mentioned in the response to reviewer 1 (point 1), we have now included the mention of biomarkers with associated references:

'Biomarkers imply that demosponges may have emerged before this, perhaps as far back as 660 Myr ago [30], although this has been disputed [31] and the debate continues [32, 33, 34].'

7. And certainly most folks do not think that oxygen rose to near-modern levels by 550 million years ago. There is evidence, for example, for a persistence of low oxygen in the oceans well into the Paleozoic, although specifics for the atmosphere are harder to constrain. There is a lot of recent literature on these stories (Lyons et al., attached, the supplement in particular, could help with some of this).

We have removed the point that the reviewer was referring to. We have changed the paragraph on the Neoproterozoic Oxidation Event with the following (see the last line in bold)

*'Towards the end of the Proterozoic, there was a further episode of increasing oxygenation known as the Neoproterozoic Oxidation Event [11, 20, 21], leading into the current Phanerozoic geological eon where oxygen levels have generally been estimated to have varied between 10% PAL and 150% PAL [11, 17, 22, 23, 24, 25] for the past 0.54 billion years, **reaching approximate modern day concentrations during the Paleozoic** [6, 26, 27].'*

8. P. 2 (Line 24): Where does the estimate of 10^{-4} come from? That is very low.

Now that we have read Lyons et al 2021 which was published after the article was submitted, we have updated the O_2 estimate to a minimum of 10^{-3} PAL during the Proterozoic.

9. P. 2 (Lines 27-29) I am confused by the assertion that 1% PAL is higher than 10^{-1} (10%) PAL. Be careful when switching between fraction and percent PAL. In general, this intro is good but not great and could be ramped up and trimmed of pieces not immediately tied to the authors' work/conclusions.

We believe that the way the sentences were worded has caused confusion. Therefore, we have changed the sentences to clarify:

'Some literature estimates suggest a larger range between 10^{-5} and 10^{-1} PAL [5, 16, 17]. However, recent one-dimensional (1D) atmospheric photochemical modelling of Earth's oxygenation history suggests geologically persistent Proterozoic oxygen levels could have been limited to values greater than or equal to 10^{-2} PAL.'

As well as our additions, we have removed some of the text that does not tie into our discussion section:

'It has been speculated that this apparent non-emergence of animal life after the GOE could be associated with two major asteroid or comet impacts ~ 2.023 and ~ 1.850 Gyr ago [34].'

And...

'Coinciding with another rise in oceanic and atmospheric oxygen [25], roughly 541 to 515 Myr ago, there was a relatively swift widespread increase in animal phyla, known as the Cambrian explosion [35]. To date, there is no conclusive single reason for the Cambrian explosion and it was likely due to several interlinked abiotic and biotic processes [35], such as the origin of biomineralisation, sea level rise [35], a change in seawater composition, and increasing oxygenation, although the data on these final two changes are not precise [36].'

And...

'If oxygen was the trigger of animal evolution, then there is a question of why animals did not evolve shortly after the GOE. However, the evolutionary origins of animals required biological functionalities which were only available after the evolution of eukaryotes [30, 31], and eukaryotes may have arisen after the GOE and the initial rise in oxygen [32, 33]. It has been speculated that this apparent non-emergence of animal life after the GOE could be associated with two major asteroid or comet impacts ~ 2.023 and ~ 1.850 Gyr ago [34].'

Our additions to the discussion (based on previous points above and those of the other reviewers) also make more of the points in the introduction relevant to our results, especially the points regarding evolution and the potential impact of UV radiation.

Response to D. S. Abbot:

1. Comparison with other work: A major conclusion of this paper is that O₃ columns are lower than previous studies have found for a given O₂ level. This result is presented in Figure 6 and page 11, as well as discussed on page 16. After reading this I was left unsatisfied about why this is the case. It's worth thinking through the specific mechanistic explanation and presenting it. The fact that WACCM6 is newer and includes more processes does not necessarily mean it is correct. The reader will have more confidence in your results if you can explain the exact improvement that leads to the difference. Here are some suggestions for working on this:

- 1) Make it more clear in Figure 6 and in the text which previous simulations are 1D and which are 3D.

We have changed the line type and the colour of the data from the 3D model used by Way et al 2017 to distinguish it from 1D models. We have also noted which data belongs to 1D models and the 3D model in the legend and the caption.

- 2) Perform 1D simulations with the WACCM6 chemical core and compare them to previous 1D work. This will allow you to separate out differences in chemical schemes from effects of going from 1D to 3D.

- 3) Perform simulations in WACCM6 with chemistry and physics decoupled. Test if this is the main reason for differences with Michael Way's ROCKE3D results, or if it is due to a different chemical scheme or differences in the radiative effect of O₂.
- 4) You point to the diurnal cycle as a key to differences with previous work. This is easy enough to test. When you have a 1D version of WACCM6 chemistry you can turn a diurnal cycle on and off and see what effect it has. You can also do this in the full 3D version, but this will be more expensive.

To respond to points 2, 3, and 4 points above, we agree with the reviewer that these are good suggestions.

However, future work would need to be conducted to attempt to isolate the exact reasons why our estimates of ozone columns are lower than prior studies and would require developing a hierarchy of model configurations to separate the interacting contributions of chemistry and resolved dynamics. For example, the chemical kinetics used in prior 1D simulations could be updated with those from the chemical scheme in WACCM. Since WACCM is a fully interactive model it is likely not feasible to constrain the photolysis to a constant zenith angle and reduce its value and not significantly impact the atmospheric dynamics, constituent transport, and thermal structure.

That being said, we have updated the manuscript to include some suggestions for future work, and to note that the possible reasons for the differences are require confirmation (see discussion section b).

'Why are our simulations predicting lower O₃ columns compared to previous work? Untangling the exact reasons and quantifying their magnitudes is difficult without a detailed model intercomparison. The following paragraph details possible reasons for discrepancies, which will require investigation with models to confirm.'

and further down in the section...

'To isolate the reasons for the differences, future work is required to test our hypotheses. This could include incorporating 3D model chemical schemes into 1D chemical schemes, and vice versa, as well as setting a constant solar zenith angle in every grid box in 3D models (multiplying photolysis rates by 0.5), and potentially adjusting heating rates for constant illumination.'

2. Methane: I am not an atmospheric chemist, but I was surprised by the result that less O₂ leads to less CH₄ (if I understood correctly). I would have assumed that increasing O₂ increases oxidation of CH₄ and would therefore decrease its equilibrium value. I think it would be good to specifically contrast your results with this intuition that many people may have. I also think it would be good to explain how this works in a bit more detail and add a schematic figure showing it. If others are naive like me about this, it could be a good way to educate the field.

We have updated the manuscript to describe this effect in more detail in the caption of Figure 8, which is a schematic representation of why CH₄ decreases when O₂ decreases. See the caption of Fig. 8:

'When the O₂ concentration reduces, the mixing ratio of CH₄ reduces. On the left is how the CH₄ loss rate varies with atmospheric pressure for the three major loss mechanisms of CH₄: photolysis, reaction with OH, and reaction with O(¹D). Shown alongside the CH₄ mixing ratio profiles for the the PI (middle) and 0.1% PAL (right) atmospheres are yellow arrows which indicate UV radiation travelling down through the atmosphere. UV radiation is attenuated by the O₃ layer. When O₂

decreases, the O_3 column abundance decreases, such that increased amounts of UV radiation penetrate into the troposphere. Through photolysis, this produces more OH (e.g. photolysis of H_2O and H_2O_2) and $O(^1D)$ molecules (e.g. photolysis of H_2O , O_2 , O_3 , and N_2O) which then react with CH_4 , decreasing its abundance. $CH_4 + h\nu$ represents photolysis of CH_4 by a photon with frequency ν , where h is Planck's constant. Note that the size of the arrows and the size of the O_3 layers do not indicate the actual magnitude of relative UV fluxes and O_3 column abundances between atmospheres, respectively.'